# OGG1 and MUTYH repair activities promote telomeric 8-oxoguanine induced senescence in human fibroblasts

Mariarosaria De Rosa[1], Ryan P. Barnes[1,5,6], Ariana C. Detwiler[1,6], Prasanth R. Nyalapatla[2], Peter Wipf [1,2] & Patricia L. Opresko [1,3,4] ✉

Telomeres are hypersensitive to the formation of the common oxidative lesion 8-oxoguanine (8oxoG), which impacts telomere stability and function. OGG1 and MUTYH glycosylases initiate base excision repair (BER) to remove 8oxoG or prevent mutation. Here, we show OGG1 loss or inhibition, or MUTYH loss, partially rescues telomeric 8oxoG-induced premature senescence and associated proinflammatory responses, while loss of both glycosylases causes a near complete rescue in human fibroblasts. Glycosylase deficiency also suppresses 8oxoG-induced telomere fragility and dysfunction, indicating that downstream single-stranded break (SSB) repair intermediates impair telomere replication. Preventing BER initiation suppresses PARylation and confers resistance to the synergistic effects of PARP inhibitors on 8oxoG-induced senescence. However, OGG1 activity is essential for preserving cell growth after chronic telomeric 8oxoG formation, whereas MUTYH promotes senescence to prevent chromosomal instability from unrepaired damage. Our studies reveal that inefficient completion of 8oxoG BER at telomeres triggers cellular senescence via SSB intermediates which disrupt telomere function.

Telomeres are nucleoprotein structures that protect the ends of linear chromosomes. In vertebrates, telomeres consist of tandem TTAGGG repeats and end in a single-stranded 3′ G-rich overhang that forms a loop structure in cooperation with shelterin, the telomere capping complex[1,2]. Telomeric DNA structure and shelterin prevent the chromosome end from being falsely recognized as a chromosome break[3]. Telomeres shorten during cell division and critically short, dysfunctional telomeres activate the canonical DNA damage response (DDR), leading to p53 activation and downstream effectors that enforce proliferative arrest and senescence, apoptosis, or autophagy depending on the cell type or context[3–5]. A few dysfunctional telomeres are sufficient to trigger senescence in human fibroblasts[6]. Dysfunctional telomeres, detected by the co-localization of telomeric DNA with DDR factors γH2AX or 53BP1, are hallmarks of senescent cells and increase in various cell types and tissues with age, and in diseases associated with oxidative stress and inflammation[7–9]. Dysfunctional telomeres have been observed even in the absence of shortening and extensive cell divisions in low proliferative cells and tissues in vivo, and are proposed to arise from irreparable breaks or damage in telomeres[9–12].

The term TelOxidation was coined to describe the hypersensitivity of telomeres to oxidative stress[13]. Oxidative stress arising from endogenous sources, including inflammation, and exogenous sources such as pollution, is not only associated with accelerated telomere shortening but also increased dysfunctional telomeres even in the absence of attrition[14]. Telomeric TTAGGG repeat sequences are preferred sites for the production of the highly common oxidative lesion 8-oxoguanine (8oxoG), which forms only on the telomere lagging strand[15–17]. Using a chemoptogenetic tool to selectively produce 8oxoG

[1]UPMC Hillman Cancer Center at the University of Pittsburgh, Pittsburgh, PA, USA. [2]Department of Chemistry, University of Pittsburgh, Pittsburgh, PA, USA. [3]Department of Pharmacology and Chemical Biology, University of Pittsburgh School of Medicine, Pittsburgh, PA, USA. [4]Department of Environmental and Occupational Health, University of Pittsburgh School of Public Health, Pittsburgh, PA, USA. [5]Present address: Department of Cancer Biology, University of Kansas Medical Center, Kansas City, KS, USA. [6]These authors contributed equally: Ryan P. Barnes, Ariana C. Detwiler. ✉e-mail: plo4@pitt.edu

at telomeres, we demonstrated that the chronic and persistent formation of telomeric 8oxoG in HeLa LT cancer cells accelerates telomere shortening and drives telomere crisis[18]. More recently, we showed that acute production of telomeric 8oxoG in non-diseased human fibroblasts and epithelial cells triggers rapid telomere dysfunction and p53-mediated cellular senescence in the absence of telomere shortening[19]. We further showed that 8oxoG induces telomere dysfunction by causing replication stress at telomeres. However, the role of 8oxoG processing and repair in telomere stability and cellular outcomes remained unknown.

Base excision repair (BER) preserves the genome after 8oxoG formation[20,21]. 8oxoG glycosylase (OGG1) specifically recognizes and removes 8oxoG opposite C[22], producing an apurinic/apyrimidinic (AP) site, which is primarily cleaved by AP endonuclease 1 (APE1), generating a nick. DNA polymerase β (Pol β) fills the gap, then DNA ligase III seals the nick. 8oxoG is prone to mispairing with A if not repaired prior to DNA replication, causing G:C to T:A transversions[23–26]. To prevent this, mutY glycosylase (MUTYH) removes the undamaged A opposite 8oxoG, then polymerase lambda inserts a C to generate an 8oxoG:C basepair which is eventually restored to G:C by OGG1 mediated repair[21,27]. Poly (ADP-ribose) polymerases PARP1 or PARP2 bind AP and nick repair intermediates and synthesize poly(ADP-ribose) PAR chains to recruit downstream BER proteins[28]. XRCC1 interacts with Pol β and ligase III, and prevents PARP1 trapping on DNA[29]. MUTYH deficiency in mice and humans increases tumorigenesis, along with 8oxoG-induced G to T mutations, which is exacerbated by additional OGG1 loss in mice[30–32]. Furthermore, UV-DDB promotes turnover of OGG1 and MUTYH, facilitating hand-off to APE1 and underscoring the importance of rapid and efficient repair progression[33,34]. While 8oxoG damage increases with age[35], potential roles for BER processing of telomeric 8oxoG in cellular aging are poorly understood.

Here we used our chemoptogenetic tool to selectively produce 8oxoG at telomeres in human fibroblasts singly or doubly deficient in OGG1 and MUTYH glycosylases. Surprisingly, we observed that knockout of OGG1 or MUTYH partially rescued multiple hallmarks of telomeric 8oxoG-induced senescence, whereas double knock-out caused a near-complete rescue. Moreover, telomeric 8oxoG induced fewer fragile telomeres in single knock-out cells, compared to wild type, and virtually no increase in double knock-out cells. Consistent with the conversion of 8oxoG to single-strand breaks (SSB) intermediates, the prevention of BER initiation in OGG1 and MUTYH deficient cells suppressed telomeric 8oxoG induced PARylation and rendered cells insensitive to synergistic effects of PARP inhibitors on damage-induced senescence. However, OGG1 is required to protect cells from senescence caused by chronic telomeric 8oxoG, whereas MUTYH promotes senescence to prevent chromosomal instability from persistent damage over time. Collectively, our studies indicate that inefficient completion of BER at telomeres triggers cellular senescence via an accumulation of repair intermediates which impair telomere replication and stability.

## Results

### OGG1 and MUTYH deficiency reduces sensitivity to acute oxidative telomere damage

Our chemoptogenetic tool generates 8oxoG by directing localized production of highly reactive singlet oxygen ($^1O_2$) at telomeres[18]. Photosensitizer di-iodinated malachite green (MG2I) dye (D) produces $^1O_2$ when bound to a fluorogen activating peptide (FAP) fused with telomeric protein TRF1, and then activated with 660 nm light (L)[36,37]. Acute telomeric 8oxoG production in human FAP-TRF1 expressing BJ hTERT fibroblasts (BJ FAP-TRF1) and primary BJ fibroblasts increases senescent cells just 4 days after dye and light (DL) exposure[19]. To determine the role of BER in telomeric 8oxoG-induced senescence, we disrupted OGG1 and MUTYH genes to generate single and double knock out (KO) cells, and confirmed similar FAP-TRF1 expression for consistent targeted 8oxoG production among the cell lines (Fig. 1A and

Supplementary Fig. 1A). While OGG1 can also remove FapyG lesions, $^1O_2$ does not induce this lesion[20,38]. Surprisingly, relative cell counts 4 days after 20 min of DL exposure showed a partial rescue of 8oxoG-induced growth reduction in single KO cells, and a near complete rescue in DKO cells, while all cell lines were unaffected by dye or light alone (Fig. 1B and Supplementary Fig. 1B). Increasing the DL exposure from 5 to 20 min amplified the differential response of glycosylase deficient and WT cells to telomeric damage (Supplementary Fig. 1C), consistent with greater 8oxoG production with longer exposures[36]. Simultaneous treatment with 20 min DL and the OGG1 inhibitor TH5487 (OGG1i), which prevents OGG1 binding[39], also partially rescued the cell growth reduction in WT cells, similar to OGG1 KO, whereas a close structural analog (OGG1i$^{NA}$) had no effect (Supplementary Fig. 1D). MUTYH KO cells treated with DL and OGG1i resembled the response of the DKO cells, yielding relative cell counts of 88 and 80%, respectively (Supplementary Fig. 1E and Fig. 1B). The DKO likely did not produce an additive effect due to different activities: only OGG1 removes 8oxoG, while MUTYH initiates BER after A misinsertion.

To investigate whether the glycosylase KO rescue of telomeric 8oxoG-induced growth reduction was specific to telomeric damage, we treated the single and double KO cells with the common oxidant potassium bromate (KBrO$_3$), which primarily induces 8oxoG lesions in DNA[40,41]. Treatment with 5 mM or 10 mM KBrO$_3$ caused a growth reduction similar to, or greater than, 20 min DL in the WT cells, respectively (Fig. 1C and Supplementary Fig. 1F). However, OGG1 KO cells showed slightly more sensitivity to 5 mM KBrO$_3$ than WT, whereas the MUTYH KO and DKO showed a reduced rescue compared to telomere-specific damage. Only the DKO cells showed rescue with 10 mM KBrO$_3$ treatment, while the single KO cells resembled WT (Supplementary Fig. 1F). The estimated average 8oxoGs produced with 5 and 10 mM KBrO$_3$ is 3300 and 6600 per cell, respectively, based on extrapolation from published 8oxoG measurements after 20–40 mM KBrO$_3$ exposures[42–44]. This far exceeds damage produced with 20 min DL at an estimated 0.8 to 2 8oxoGs per telomere; 76-184 total telomeric 8oxoGs per cell based on the average BJ-hTERT telomere length (~10 kb), and previous measurements[45]. These data suggest that in addition to telomeres being hypersensitive to 8oxoG formation, they are also more sensitive to possible deleterious effects of glycosylase activity, compared to elsewhere in the bulk genome.

Next, we asked whether the differential growth reduction after telomere-specific damage among the cell lines was due to differences in the induction of senescence[46]. In contrast to WT cells, which showed a dramatic 4-fold increase in senescence-associated β-galactosidase (SA-β-gal) staining 4 days after 20 min DL exposure, the OGG1 and MUTYH KO cells showed a lesser (about 2-fold) increase, and the DKO cells remained largely unaffected (Fig. 1D, E). However, glycosylase deficiency failed to rescue senescence induced with 5 or 10 mM KBrO$_3$ treatment (Fig. 1F), confirming the enhanced sensitivity of telomeric regions to glycosylase activity compared to the bulk genome. Consistent with morphological changes of senescent cells[47,48], the nuclear area of WT cells increased significantly 4 days after DL treatment, whereas the glycosylase deficient cells showed minimal changes (Fig. 1G). Finally, since glycosylase deficiency alone reduced cell growth rates (Supplementary Fig. 1G), we asked if this contributed to the reduced sensitivity to telomeric 8oxoG. We observed that culturing WT cells in limiting serum slowed their growth, but did not rescue the 8oxoG-induced decrease in relative cell number and increase in SA-β-gal staining (Supplementary Fig. 1H, I). Taken together, our data show that OGG1 or MUTYH deficiency in non-diseased cells reduces sensitivity to the acute telomeric 8oxoG induction of senescence-associated phenotypes.

### Glycosylase activity enhances telomeric 8oxoG-induced cytoplasmic DNA and proinflammatory response

Another hallmark of senescent cells is the appearance of cytoplasmic chromatin fragments (CCFs), of which we showed an increase in BJ

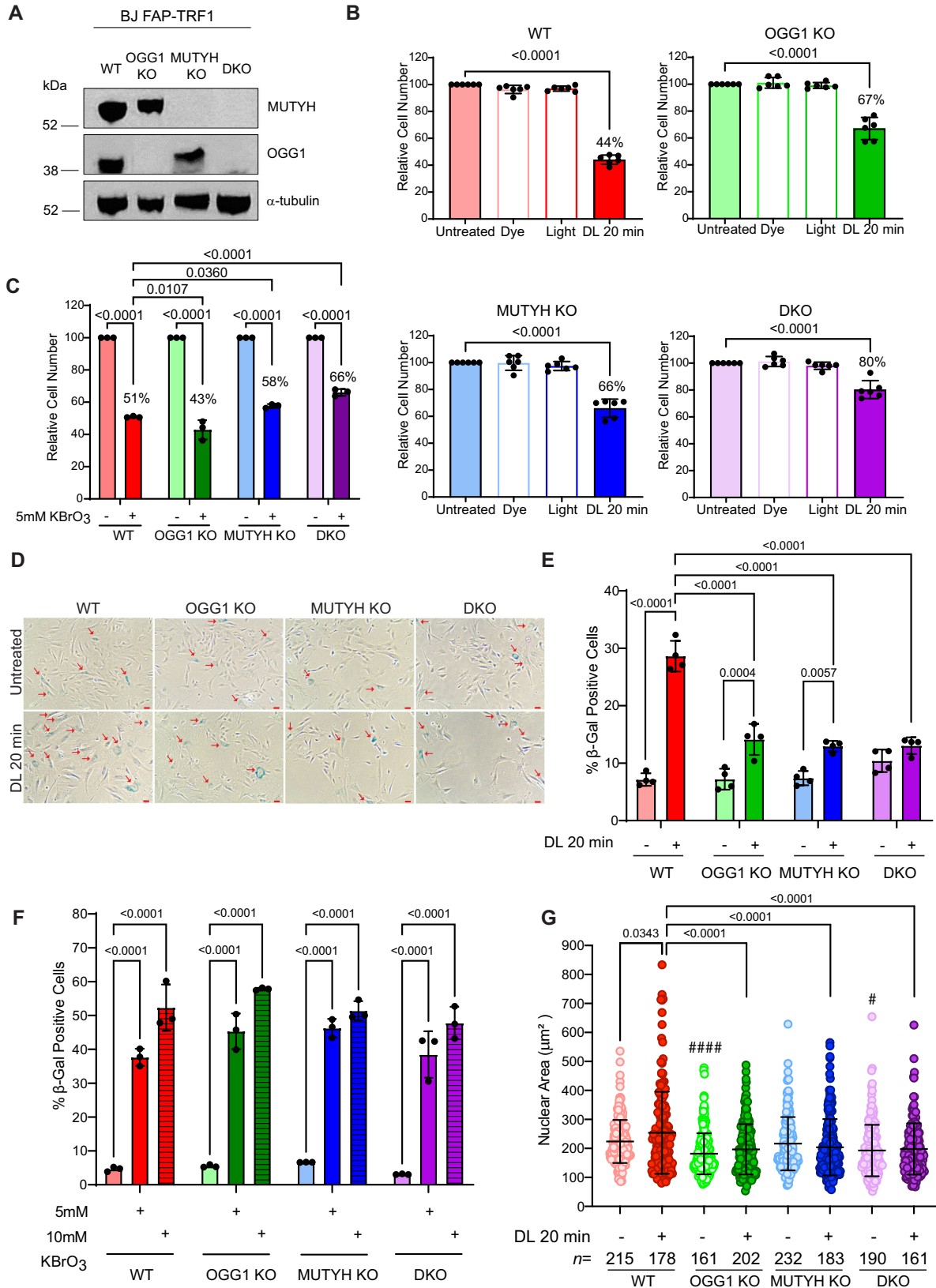

FAP-TRF1 cells after telomeric 8oxoG damage[19]. CCFs resemble micronuclei (MN) but arise outside of mitosis by apparent blebbing of the chromatin into the cytoplasm[49]. As further confirmation that the repair deficient cells have an attenuated response to telomeric 8oxoG damage, CCF production in the single KO and DKO cells was reduced or abrogated, respectively, 24 h and 4 days after DL (Fig. 2A, B and

Supplementary Fig. 2A). Damage-induced CCFs levels are similar in the single KO and DKO cells, but are not higher in DKO compared to untreated cells after 24 h (Fig. 2B). The percent of CCFs containing telomeric DNA was comparable in the glycosylase proficient and deficient cells (Supplementary Fig. 2B). In addition, the proportion of CCFs containing γH2AX or 53BP1 was similar before or after damage in

**Fig. 1 | OGG1 and MUTYH deficiency reduces sensitivity to acute oxidative telomere damage. A** Immunoblot of MUTYH and OGG1 in parental BJ FAP-TRF1 and MUTYH KO, OGG1 KO or DKO cell lines. α-tubulin was used as a loading control. **B** Cell counts of BJ FAP-TRF1 cell lines obtained 4 days after recovery from 20 min dye + light (DL) treatment, relative to untreated cells. Data are mean ± SD from six independent experiments; *P*-values were obtained using two-way ANOVA. **C** Cell counts of BJ FAP-TRF1 cell lines were obtained 4 days after recovery from 1 h treatment with 5 mM KBrO$_3$, relative to untreated cells. Data are mean ± SD from three independent experiments; *P*-values were obtained using two-way ANOVA. **D** Representative images of β-galactosidase positive cells obtained 4 days after recovery from no treatment or 20 min DL treatment. Red arrows mark positive

cells. Scale bars = 100 µm. **E** Percent β-galactosidase positive cells. Data are mean ± SD from four independent experiments; *P*-values were obtained using two-way ANOVA. **F** Percent β-galactosidase positive cells following treatment with the indicated KBrO$_3$ concentrations. Data are mean ± SD from three independent experiments; *P*-values were obtained using two-way ANOVA. **G** Size of nuclear area (µm$^2$) of cells obtained 4 days after recovery from no treatment or 20 min DL. Data are mean ± SD from the indicated *n* number of nuclei analyzed from three independent experiments; each dot represents a nucleus. *P*-values were obtained using ordinary one-way ANOVA; comparison of untreated cell lines with untreated WT indicated by #*p* = 0.0194; ####*p* < 0.0001. For all panels, only comparisons yielding significant *P*-values are shown. Source data are provided as a Source Data file.

all cell lines, but the majority were γH2AX+ and 53BP1- (Fig. 2C). This is consistent with the cytoplasmic DNA species arising from chromatin blebbing outside of mitosis as we reported previously for WT cells[19]. In agreement, the oxidative telomere damage did not increase chromatin bridges in any of the cell lines (Fig. 2D). This suggests that while the damage-induced cytoplasmic DNA species were decreased in glycosylase-deficient cells, the mechanism of production was the same as in WT cells. Since MN or CCFs can arise from DNA breaks related to apoptosis induction, we tested for apoptosis by Annexin V (AV) and propidium iodide (PI) staining. Our results confirmed that 20 min DL exposure did not induce apoptosis in any of the cell lines, unlike the UV-treated controls (Supplementary Fig. 2C, D).

Senescent cells secrete a large array of proinflammatory factors, called the senescence-associated secretory phenotype (SASP), which is mainly driven by the CCFs triggering the cytoplasmic DNA sensor cyclic GMP-AMP synthase (cGAS)[49,50]. Targeted telomeric oxidative damage induced both cGAS and SASP in repair-proficient cells[19], as well as phosphorylation of cGAS effector STING (Supplementary Fig. 2E). Therefore, we investigated whether these phenotypes were rescued in the glycosylase deficient cells. In response to DL, the repair deficient cells showed significantly fewer cGAS+ CCFs (Fig. 2E, F). Glycosylase-deficient cells also displayed decreased secretion of various pro-inflammatory cytokines and chemokines, compared to WT (Fig. 2G). DL exposure increased all factors above background in WT cells except IL-6, a few in single knock out cells, but none in DKO cells (Fig. 2G and Supplementary Fig. 2F). Curiously, DKO cells showed higher basal levels of three factors, but these were not increased further with DL exposure (Supplementary Fig. 2F). In summary, cells lacking OGG1 or MUTYH, or both, show reduced formation of telomeric 8oxoG induced senescence-associated and inflammation-promoting cytoplasmic DNA species.

## Glycosylase deficiency suppresses telomeric 8oxoG-induced replication stress

Telomeric 8oxoG induces senescence by impairing telomere replication, as indicated by increased fragile telomeres and a localized DNA damage response (DDR) that ultimately activates p53-mediated premature senescence[19]. We predicted that the greater senescence induction in WT cells compared to OGG1 and MUTYH deficient cells, was due to the glycosylases enhancing or provoking telomeric replication stress and subsequent DDR activation. To test this, we examined telomere integrity by fluorescent in situ hybridization (FISH) on metaphase chromosomes, and scored the chromatid ends as normal (one distinct telomeric foci), signal-free ends (no staining) or fragile telomeres (multiple foci) (Fig. 3A, B and Supplementary Fig. 3A). Transient depletion of p53 through siRNA (Supplementary Fig. 3B) allowed the damaged cells to progress into mitosis. While DL treatment did not significantly increase signal-free ends in any of the cell lines (Fig. 3A), telomeric 8oxoG induced fewer fragile telomeres in the single KO cells compared to WT, and virtually no increase in the DKO cells (Fig. 3B). These results are consistent with the attenuated senescent phenotypes observed in the glycosylase deficient cells after damage, and suggest that OGG1 and MUTYH deficiency suppresses replication stress induced by telomeric 8oxoG.

Telomeric replication stress and fragility lead to DDR signaling and activation of ATR and ATM kinases at the telomeres, which can trigger senescence if not resolved[19,51]. Therefore, we tested whether glycosylase deficiency also suppresses activation of the ATR/Chk1 and ATM/Chk2 pathways and downstream induction of p53 and p21, after telomeric 8oxoG damage. Phosphorylation of Chk1 after 8oxoG damage was attenuated in single knockout and DKO cells, compared to WT, although it was higher in MUTYH KO cells compared to the other glycosylase-deficient cells (Fig. 3C). These results are consistent with less replication stress after telomere damage in the glycosylase-deficient cells compared to WT. While both OGG1 and MUTYH KO cells showed ATM/Chk2, p53, and p21 activation shortly (3 h) after 20 min of DL, similar to WT cells, the DKO cells showed an attenuated response (Supplementary Fig. 3C–E). As a positive control, we confirmed that all the cell lines mount a robust DDR after 20 J/m$^2$ UVC exposure (Supplementary Fig. 3F). Thus, while lesion processing by either OGG1 or MUTYH can activate ATM or ATR signaling, the impaired DDR activation in the DKO cells suggests the BER initiation at telomeric 8oxoG lesions contributes to ATM/ATR kinase activation. Analysis of localized telomeric DDR confirmed increased γH2AX and/or 53BP1 recruitment to telomeres 24 h after 8oxoG damage in WT cells, which was attenuated in single KO cells and suppressed in DKO cells (Fig. 3D, E and Supplementary Fig. 3G, H). In summary, the loss of both MUTYH and OGG1 glycosylases suppresses telomeric 8oxoG induction of DDR+ telomeres, DDR signaling, and p53 activation, consistent with the attenuated production of senescent cells.

## BER intermediates promote telomeric 8oxoG-induced senescence

Next, we examined the mechanism by which glycosylase deficiency attenuates the telomeric 8oxoG-induced DDR and senescence. We hypothesized that BER initiation by OGG1 and MUTYH and the consequent formation of abasic sites and downstream SSBs may impair telomere replication and trigger cellular senescence. To test for SSB intermediates after telomeric 8oxoG production, we adopted the microscopy-based exo-FISH method in which SSBs arising from 8oxoG incision provide substrates for exonuclease III digestion at 3' ends to reveal single-stranded DNA for hybridization with (TTAGGG)$_3$ telomeric probes[52] (Fig. 4A). 20 min DL followed by 30 min recovery significantly increased exo-FISH signal intensity to levels comparable to 5 mM KBrO$_3$ treatment in WT cells, but not in OGG1 KO and DKO cells (Fig. 4B, C and Supplementary Fig. 4A). Interestingly, MUTYH KO cells showed reduced induction of exo-FISH intensity compared to WT cells, consistent with some evidence that MUTYH may enhance OGG1-mediated 8oxoG repair[31]. Using the (CCCTAA)$_3$ telomeric probes to reveal SSBs arising from MUTYH-initiated BER revealed a slight increase in exo-FISH signal intensity in the WT and OGG1 KO cells, barely above background (Supplementary Fig. 4B–D). This suggests that 30 min after the damage is insufficient to allow significant adenine misincorporation opposite 8oxoG to detect MUTYH BER activity. We then examined Poly-ADP ribosylation (PARylation) as an indirect indicator of repair intermediates. PARP1 and PARP2 are recruited to SSBs and PARylate themselves and other repair proteins to promote BER completion[28,53]. We added Poly(ADP-Ribose) Glycohydrolase

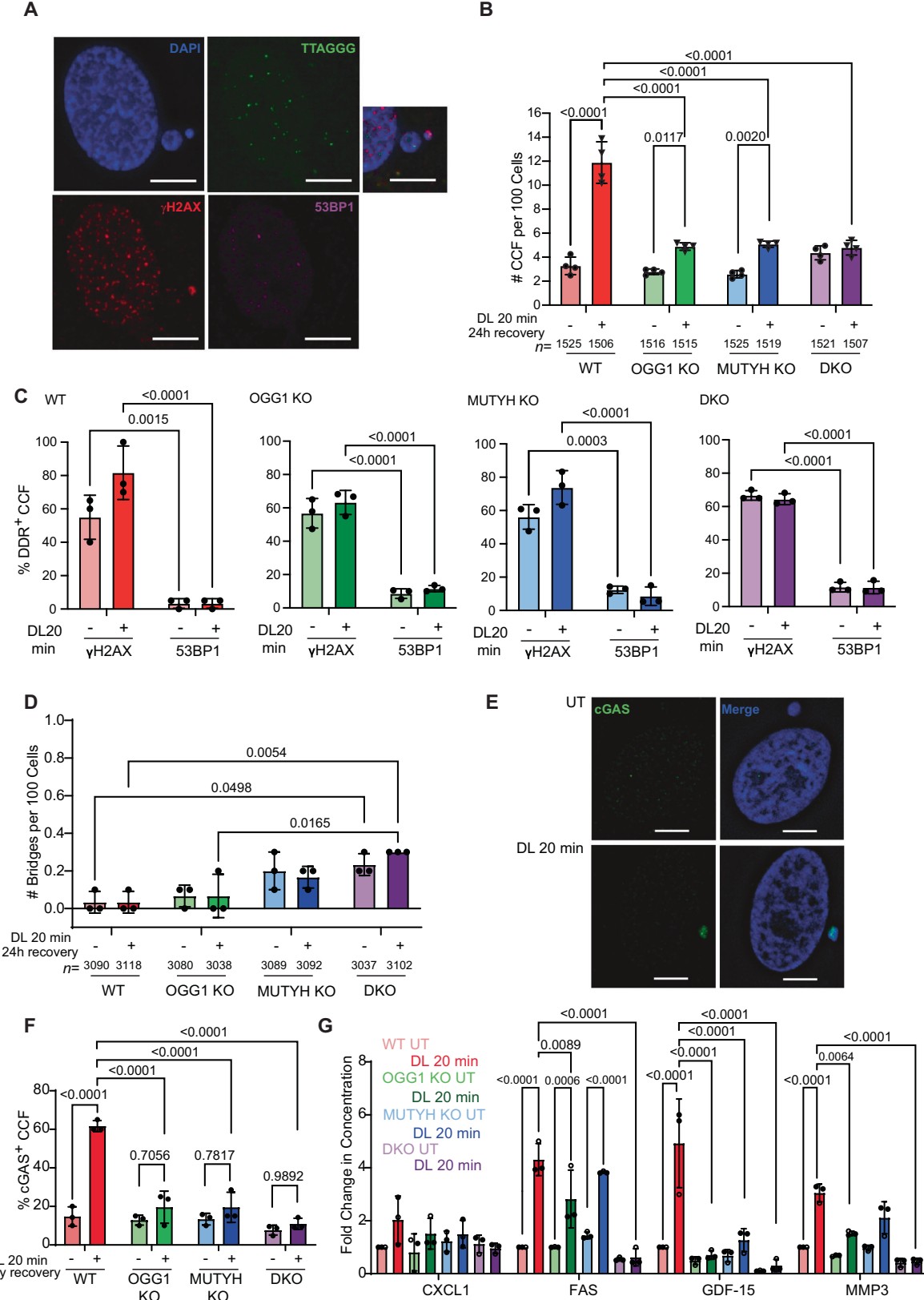

inhibitor (PARGi) PDD00017272 during the DL exposure to preserve the PAR chains[54]. We observed weak, but detectable, PARylation after 5 min DL, and much more robust PARylation after 20 min DL, which is abolished with PARP inhibitor Olaparib (Supplementary Fig. 4E). Dye or light-only controls failed to induce PARylation, confirming that PARP1/2 activity depends on the targeted 8oxoG formation at telomeres (Supplementary Fig. 4F). Remarkably, while damage-induced PARylation was attenuated in single KO cells, compared to WT, it was greatly suppressed in DKO cells (Fig. 4D, E). These data indicate the formation of BER intermediates in the DKO cells is negligible, and importantly, confirm that the DL treatment does not directly produce SSBs as the initial lesion. Furthermore, increased PARylation

**Fig. 2 | Glycosylase activity enhances telomeric 8oxoG-induced cytoplasmic DNA and proinflammatory response. A** Representative IF-FISH image of cytoplasmic DNA species stained by DAPI (blue) containing telomeres (green), γH2AX (red), or 53BP1 (purple) in WT cells 24 h after recovery from 20 min DL. Scale bars = 10 μm. **B** Quantification of cytoplasmic chromatin fragments (CCF) 24 h after recovery from 20 min DL from (**A**). Data are mean ± SD from four independent experiments; *n* = total number of cells analyzed; *P*-values were obtained using two-way ANOVA. **C** Quantification of the percent of CCFs from (**A**) positive for DDR markers γH2AX or 53BP1 for the indicated cell lines. At least 20 CCFs were analyzed for each experiment. Data are mean ± SD from three independent experiments; *P*-values were obtained using two-way ANOVA. **D** Quantification of the percent of chromatin bridges 24 h after recovery from 20 min DL. Data are mean ± SD from

three independent experiments; *n* = total number of cells analyzed. *P*-values were obtained using two-way ANOVA. **E** BJ FAP-TRF1 WT cells stained for cGAS 4 days after 20 min DL treatment. Scale bars = 10 μm. **F** Quantification of the percent of CCFs that are cGAS positive 4 days after 20 min DL treatment as in (**C**). *P*-values were obtained using two-way ANOVA. **G** SASP analysis of BJ FAP-TRF1 cells 7 days post-treatment with DL 20 min. Concentration normalized to the final cell number in each sample. Data are presented as fold changes. Data are mean ± SD from two technical replicates of one experiment (dark circles) and one replicate from an independent experiment (open circles). *P*-values were obtained using two-way ANOVA. For all panels, only comparisons yielding significant *P*-values are shown, except for non-significant comparisons shown in **F** for clarity. Source data are provided as a Source Data file.

after 20 min DL, compared to 5 min, in WT cells is consistent with the greater difference in growth inhibition between the WT and repair-deficient cells after 20 min DL compared to 5 min (compare Fig. 1B and Supplementary Fig. 1C). Finally, APE1 depletion with targeting siRNAs also partially rescued senescence caused by 20 min DL, similar to siRNAs targeting OGG1 or MUTHY (Supplementary Fig. 4G–I). Since APE1 loss prevents the conversion of the abasic sites to an SSB, this suggests that SSB repair intermediates at telomeres may be more detrimental than abasic sites, leading to increased senescence.

Next, we reasoned that if repair intermediates were present after 8oxoG damage, then the addition of the PARP inhibitor Olaparib should increase PARP1 or PARP2 retention at these sites, and further exacerbate replication stress and senescence induction[55]. By enhancing PARP1/2 retention at SSBs, Olaparib causes aborted BER and accumulation of toxic repair intermediates[55,56]. Combined treatment of DL and Olaparib in BER proficient cells synergistically increased senescence induction, indicated by beta-gal staining, compared to DL alone. Conversely, single knock out cells were less affected, and DKO cells were insensitive to adding the PARPi with damage (Fig. 4F). Taken together, these results suggest that the initiation of BER and consequent formation of repair intermediates are drivers of telomeric 8oxoG-induced senescence.

## Both BER and replication stress activate PARylation and DDR after 8oxoG damage

While SSBs arising from BER can activate both DDR and PARylation, we previously demonstrated that replication contributes to telomeric 8oxoG-induced DDR in WT cells[19], and other studies revealed PARylation of proteins at sites of replication stress[57]. To delineate the contribution of BER and/or replication stress to 8oxoG-induced damage responses, we damaged telomeres in unsynchronized (replicating) cells, and quiescent (non-replicating) cells (Fig. 5A). Cells were synchronized to G0/G1 by serum starvation and contact inhibition, as confirmed by the lack of cyclin A and EdU staining 48 h after entering quiescence (Supplementary Fig. 5A, B). We observed damage-induced Chk2 phosphorylation in both unsynchronized and quiescent WT cells, indicating that telomeric 8oxoG processing by BER can trigger the DDR (Supplementary Fig. 5A). However, telomeric 8oxoG damage induced Chk1 phosphorylation only in unsynchronized cells, consistent with replication stress signaling (Supplementary Fig. 5A). Telomeric 8oxoG also induced PARylation in quiescent cells, although reduced compared to replicating cells (Fig. 5B, C). IF-teloFISH revealed that telomeric 8oxoG generated PAR enrichment specifically at telomeres, which was attenuated in quiescent cells compared to unsynchronized cells (Fig. 5D, E). These data indicate that replication stress induced by 8oxoG and its repair intermediates enhances PARylation, consistent with a role for PARPs in replication stress[58].

## OGG1 loss sensitizes cells to chronic telomere damage while MUTYH loss promotes resistance

Here we show OGG1 or MUTYH glycosylase deficiency can partially suppress the deleterious effects of acute oxidative telomere damage in

fibroblasts. However, we found previously that OGG1 repair of telomeric 8oxoG is critical for telomere stability in cancer cells after chronic telomere damage[18]. Therefore, we investigated how repeated exposure to targeted 8oxoG impacted telomere stability and cellular growth in the BJ FAP-TRF1 fibroblasts. To mimic chronic oxidative stress at telomeres, we treated cells with DL each day, except every fourth day when cells were harvested for analysis, over 24 days for 18 treatments (N18) (Fig. 6A). Interestingly, loss of MUTYH or OGG1 alone reduced growth compared to WT cells, but the DKO cells grew nearly as well as WT (Fig. 6B). The WT and OGG1 KO cells were the most severely affected by repeated DL exposures as indicated by the shallow slopes of their growth curves, and near plateau for the OGG1 KO, while the MUTYH KO cells showed reduced growth but surpassed the WT cells over time (Fig. 6B, C). Although chronic telomere damage significantly decreased the PD of DKO cells, surprisingly, the cells continued growing without reaching a plateau (Fig. 6B). Consistent with these results, the significant increase in the percent of β-gal positive WT and OGG1 KO cells caused by chronic damage, was attenuated in the MUTYH KO and DKO cells (Fig. 6D). Chronic damage also increased apoptosis in MUTYH proficient cells, consistent with this glycosylase's role in inducing cell death under oxidative stress (Supplementary Fig. 6A)[30]. This suggests the loss of MUTYH, but not OGG1, partially rescues senescence and prevents cell death caused by chronic oxidative damage at telomeres, leading to reduced but sustained growth.

Next, we examined telomere integrity after chronic oxidative telomere damage. The exo-FISH assay conducted after the last 20 min DL exposure confirmed that chronic damage significantly increased telomeric SSB intermediates in WT cells, but not in DKO cells (Fig. 6E–G). Pretreatment with formamidopyrimidine DNA glycosylase (FPG), which removes 8oxoG and cleaves the DNA backbone, reveals both 8oxoG and SSBs. After chronic damage, the increased exo-FISH signal intensity with FPG pre-treatment represented a combination of 8oxoG and SSBs for WT cells, but only 8oxoG for DKO cells since these cells did not show a significant increase in exo-FISH without FPG treatment (Fig. 6F, G). In contrast to acute damage, chronic damage significantly increased signal free ends, suggestive of telomere losses, in WT and OGG1 KO (Fig. 6H), and increased fragile telomeres in WT and OGG1 KO cells, but not in MUTYH KO and DKO cells (Fig. 6I). Furthermore, only WT and OGG1 KO cells showed a significant increase in DDR+ telomeres 24 h after the last DL (N18) (Supplementary Fig. 6B), suggesting MUTYH contributes to DDR signaling in response to oxidative damage at telomeres.

## Chronic telomere damage drives chromosomal instability in OGG1 and MUTYH doubly deficient cells

Consistent with damage-induced telomere instability, 18 DL exposures significantly increased cytoplasmic DNA species in WT and OGG1 KO cells, but not in MUTYH KO cells (Fig. 7A). Interestingly, while the DKO cells showed less β-gal staining indicative of senescence, they showed the highest level of cytoplasmic DNA basally and after chronic damage, although WT and OGG1 KO cells showed the greatest damage-induced

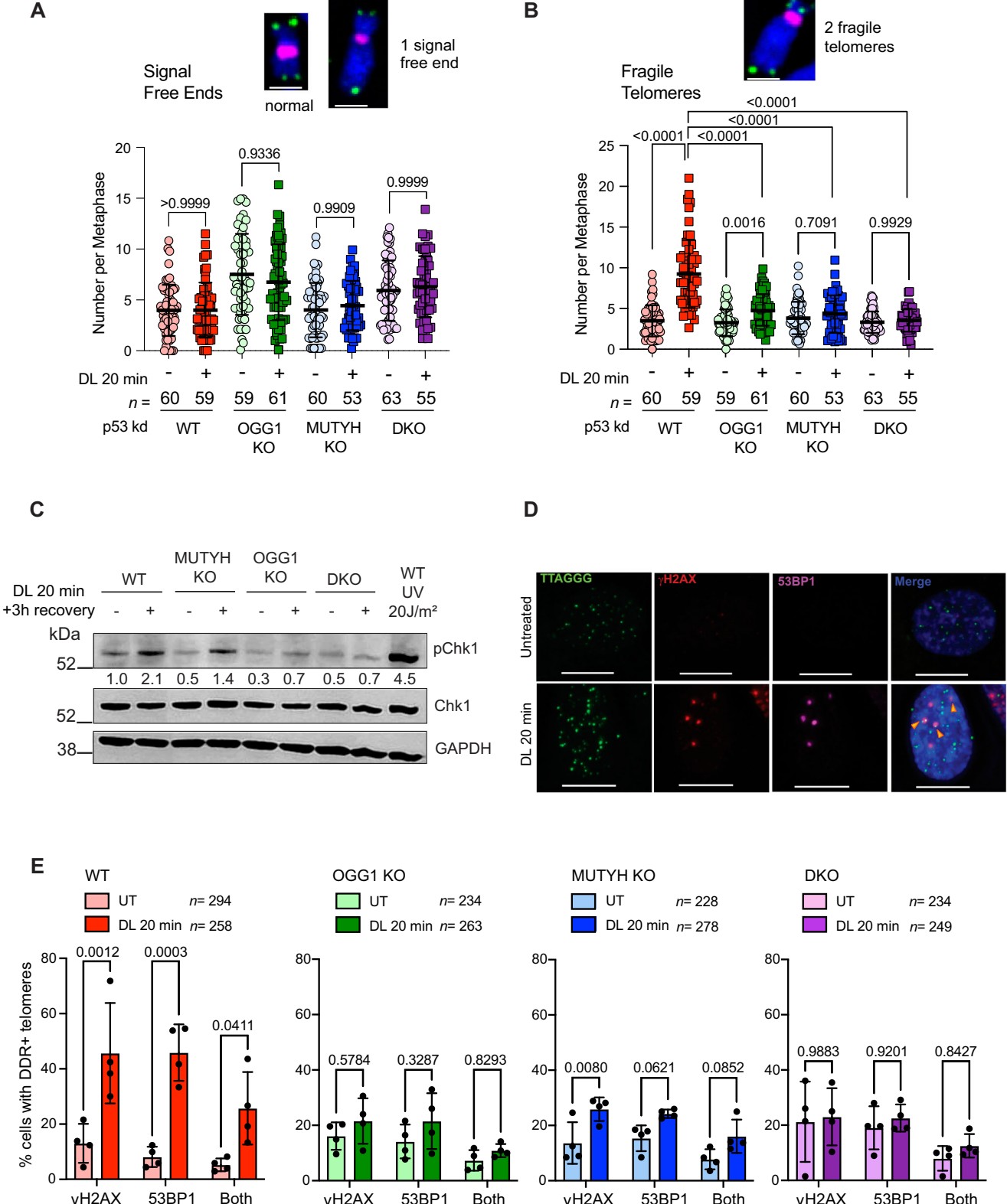

increase (Fig. 7A). Futhermore, the DKO cells exhibited increased chromatin bridges 24 h after N18, which was not detected in the WT or single KO cells, or after acute damage (Fig. 7B, C). This suggests that some cytoplasmic DNA species in DKO cells arose from chromatin bridge formation and breakage rather than senescence-associated blebbing from the nucleus. The DKO cells showed a higher percent of cytoplasmic DNA positive for 53BP1, compared to WT or single KO

cells, consistent with micronuclei arising from chromosome breaks during mitosis (Supplementary Fig. 7A, B). Remarkably, DKO cells show reduced p53 after culturing for 24 days, compared to WT and single KO cell lines, and further reduction after chronic telomere damage (Supplementary Fig. 7C). However, 20 J/m² UVC activates p53 expression in DKO cells on day 24 of culturing (Supplementary Fig. 7D), confirming these cells are proficient for p53. Control p53 KO

**Fig. 3 | Glycosylase deficiency suppresses telomeric 8oxoG-induced replication stress.** Number of telomeric signal-free chromatid ends (**A**) or fragile telomeres (**B**) per metaphase in cell lines transiently depleted for p53. Inset shows representative images of metaphase chromosomes scored as normal, signal-free end or fragile by telo-FISH (green); pink = centromeres. Scale bars = 2.5 µm. Data represent mean ± SD from the indicated $n$ number of metaphases analyzed from three independent experiments, normalized to the chromosome number; $P$-values were obtained using two-way ANOVA. **C** Immunoblot of indicated proteins in cells untreated or treated with 20 min DL, or 20 J/m$^2$ UVC light as a positive control (WT only), and recovered 3 h; pChk1 indicates phosphorylated forms; GAPDH used as a loading control. Numbers below the pChk1 blot represent normalized protein expression. **D** Representative IF images showing γH2AX (red) and 53BP1 (purple) staining with telomeres (green) by telo-FISH for WT cells 24 h after no treatment or 20 min DL. Yellow arrowheads point to NIS-Elements-defined intersections between 53BP1 and/or γH2AX with telomeres. Scale bars = 10 µm. **E** Quantification of the percent of cells showing ≥1 telomere foci co-localized with γH2AX, 53BP1, or both 24 h after no treatment or 20 min DL. Data represent the mean ± SD from the indicated $n$ number of nuclei analyzed from four independent experiments; $P$-values were obtained using two-way ANOVA. Source data are provided as a Source Data file.

cells fail to show p53 expression after chronic telomeric 8oxoG and UVC exposure (Supplementary Fig. 7C, D). These results implicate SSBs and MUTYH in telomeric 8oxoG-induced p53 activation. Chronic telomere damage impairs growth of DKO cells similarly to p53 KO cells, confirming a role for p53 (Supplementary Fig. 7E, F). However, DKO cells show more damage-induced cytoplasmic DNA species, compared to p53 KO cells (Fig. 7A and Supplementary Fig. 7G). Taken together, our results indicate that while OGG1 deficiency partially protects against acute telomeric 8oxoG, it becomes detrimental after long-term damage. Moreover, MUTYH deficiency promotes resistance to chronic telomeric 8oxoG, while rescuing senescence and cell death induction and further promoting genome instability in cells simultaneously deficient for OGG1.

## Discussion

Telomere oxidative damage has a direct role in cellular aging, as a single induction of telomeric 8oxoG is sufficient to induce cellular senescence in non-diseased cells[19]. Here, we demonstrate that OGG1 and MUTYH glycosylase activity triggers senescence via an accumulation of repair intermediates, which impair telomere replication and stability. Using our chemoptogenetic tool to selectively generate 8oxoG lesions at telomeres in human fibroblasts singly or doubly deficient in OGG1 and MUTYH glycosylases, we observed a partial or near-complete rescue, respectively, of multiple hallmarks of damage-induced senescence including the proinflammatory SASP. Our data is consistent with a mechanistic model (Fig. 7D) in which the production of SSB intermediates during repair contributes to replication stress and DDR activation at telomeres, leading to p53-mediated senescence. We uncovered a synergistic effect of telomeric 8oxoG damage and PARP inhibition in triggering cellular senescence, to which glycosylase-deficient cells are insensitive. However, chronic telomeric 8oxoG production revealed divergent roles for OGG1 and MUTYH in suppressing or promoting, respectively, damage-induced senescence or apoptosis (Fig. 7D). Our data suggest that glycosylase-produced SSBs and possible MUTYH roles at accumulated telomeric 8oxoGs in OGG1 deficient cells, drive p53 activation and thereby promote senescence.

Our findings on 8oxoG processing are consistent with evidence that excessive glycosylase activity and unbalanced BER activity can be detrimental[59]. Alkyladenine DNA glycosylase (AAG)-initiated BER can exacerbate alkylation-induced cytotoxicity and specific tissue damage in mice by causing PARP1 overactivation[60–62]. Similarly, evidence that OGG1-initiated BER can be detrimental includes reports that OGG1 loss reduces cytotoxicity and DNA breaks caused by 8-oxodGTP in A549 cancer cells[63], H$_2$O$_2$-induced PARP1 overactivation and DNA breaks in MEFs[64] and H$_2$O$_2$-induced telomere breaks in HeLa cells[65]. While these studies were limited to genome-wide damage, they indicate that cell type and context influence the impact of glycosylase-induced repair intermediates. Here, with our chemoptogenetic tool, we now show that 8oxoG production exclusively at the telomeres is sufficient to provoke robust PARylation, and that loss of both OGG1 and MUTYH in human fibroblasts is required to suppress the consequent SSB formation and PARylation, as well as sensitivity to PARPi Olaparib. This contrasts findings that AAG loss alone is sufficient to suppress PARylation from alkylation DNA damage[66]. Our result that OGG1 loss

further sensitizes cells to genome-wide 8oxoG damage, but confers partial resistance to telomere-specific damage, indicates that telomeres may be particularly sensitive to SSBs or that repair may be slower at telomeres allowing for SSB accumulation. These findings raise the possibility that the combined use of conventional chemotherapeutic agents, which cause oxidative stress in normal tissues[67], and PARP inhibitors, may contribute to premature cellular senescence and aging in cancer patients through a telomere-mediated mechanism.

Previous studies reported that enzymatic activity from AAG (also called MPG) and uracil DNA glycosylase 2 (UNG2) activate ATM/Chk2 by cooperating with APE1 endonuclease to convert glycosylase-specific lesions into SSB intermediates[68,69]. AAG and APE1 activity following MMS treatment induce PARylation and ATM activation in both replicating and quiescent cells, suggesting DDR activation can arise from SSB conversion to DSBs during replication or directly from SSB formation, respectively[69]. In these studies, the use of H$_2$O$_2$ to induce oxidative lesions was confounded by findings that oxidative stress can also activate ATM via direct protein oxidation[70,71]. In our study, we show that in the absence of general oxidative stress, the processing of 8oxoG at telomeres can activate the ATM/Chk2 pathway even in non-replicating cells. This DDR activation might arise from glycosylase excision, or from downstream repair intermediates and PARP1 recruitment. Interestingly, since Gs only exist on the telomere lagging strand, OGG1 excision should occur on one strand in non-replicating cells, ruling out the possibility that excision on opposite strands produces a DSB that activates ATM[72]. By suppressing possible DSB formation from replication fork collision with an SSB, or from excision of clustered bases on opposing strands, our data strongly suggest that SSBs can directly activate ATM kinase and DDR, at least in the context of telomeres.

Among the most surprising results of this study are the similar partial rescue of the acute damage-induced senescence phenotypes when either OGG1 or MUTYH is depleted, and the requirement to deplete both glycosylases for near-complete rescue. MUTYH excises A misinserted opposite 8oxoG after replication to prevent the accumulation of G:C→T:A transversion mutations[73]. Thus, we predicted that MUTYH knock out would cause a weaker rescue of cellular senescence after acute telomeric 8oxoG damage compared to OGG1 deficiency. However, we observed a similar rescue of all the analyzed phenotypes, and a greater effect when both glycosylases were lost. Interestingly, the mutation G396D located in the MUTYH 8oxoG recognition domain is among the most prevalent in MUTYH-associated polyposis (MAP) cancers[73], which suggests that the recognition of 8oxoG itself is a crucial MUTYH property. Consistent with telomeric 8oxoG-induced PARylation in OGG1 KO cells, others reported rapid MUTYH-mediated PARylation, and increased 8oxoG levels in MUTYH deficient cells, shortly after oxidative damage[74,75]. MUTYH can bind unproductively to 8oxoG:C lesions[31], thereby possibly provoking replication fork stalling and PARylation[57], which may partly explain why MUTYH deficient cells are less sensitive to acute telomere damage than wild-type cells. Finally, MUTYH can induce futile BER cycles during oxidative stress, because 8oxoG accumulates in the template strand and DNA polymerases can reinsert A opposite 8oxoG[30]. This may explain why MUTYH-deficient cells escape cell death caused by both acute and chronic oxidative telomeric damage.

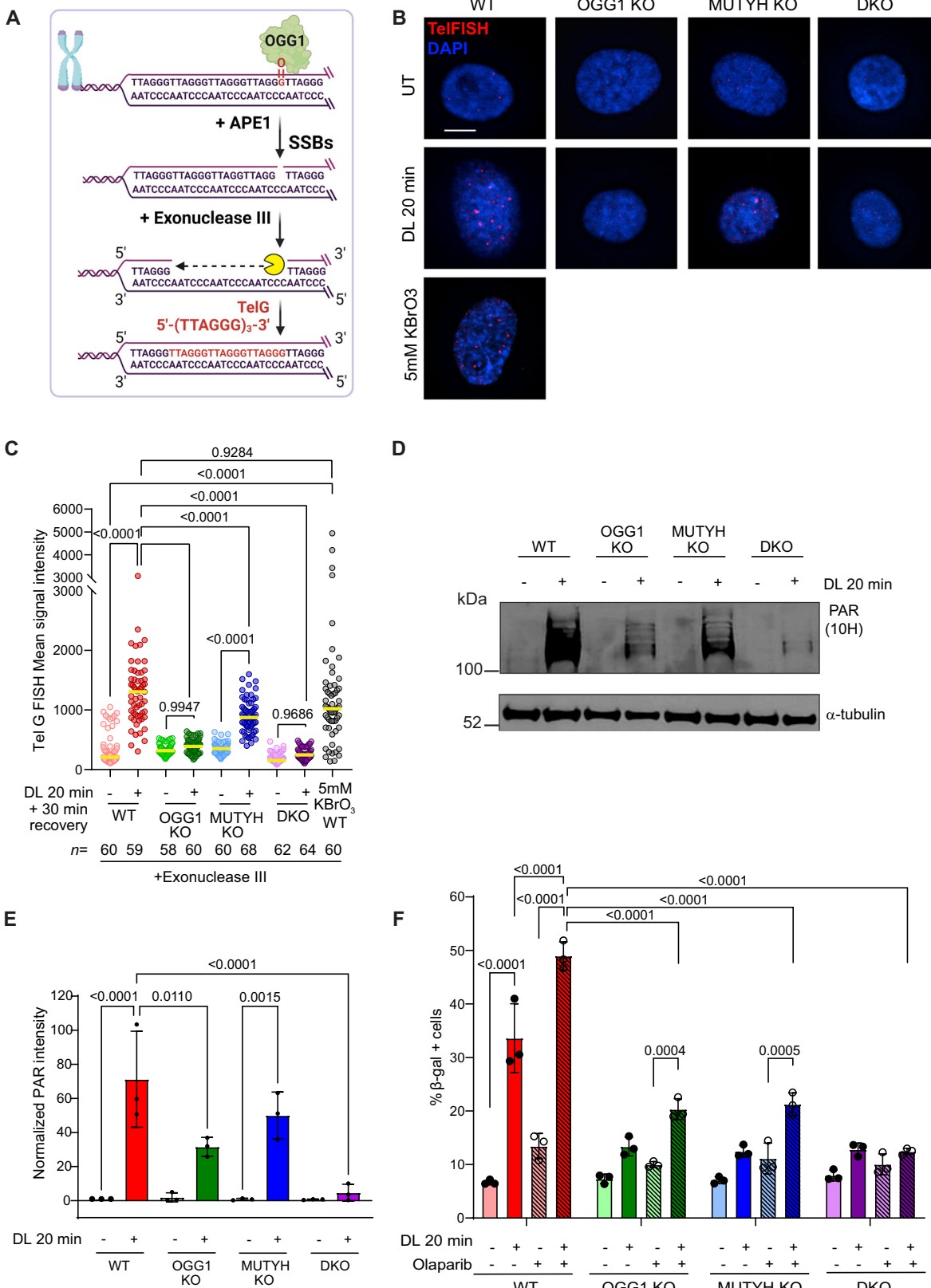

While loss of glycosylase activity provided a growth advantage after acute telomere damage, compared to repair proficient cells, OGG1 loss during the long-term telomeric 8oxoG production was highly detrimental, whereas MUTYH loss was not. Similarly, OGG1 loss exacerbated chronic telomeric 8oxoG-induced telomere losses and impaired cell growth in HeLa cells[18]. Here we found that human fibroblasts are much more sensitive to chronic telomeric 8oxoG than HeLa cells, primarily due to p53 activation. Remarkably, the additional loss of MUTYH in cells lacking OGG1 partially rescued the severe growth impairment, senescence, and p53 activation after chronic telomere damage. Our data suggests repair intermediates increase p53 expression because SSBs are suppressed in doubly deficient MUTYH

**Fig. 4 | BER intermediates promote telomeric 8oxoG-induced senescence.**
**A** Schematic of the detection of SSB repair intermediates from 8oxoG processing at telomeres using exo-FISH. Created in BioRender. De Rosa, M. (2024) https://BioRender.com/y55q493. **B** Representative images of exo-FISH performed in cells untreated, treated with 20 min DL or 5 mM KBrO$_3$ (WT only), showing telomeres harboring SSBs on the G-rich strand (red foci). Cells were treated with exonuclease III before hybridization with TelG FISH probe. Scale bar = 10 μm. **C** Quantification of exo-FISH signal intensity from **B**. Each data point represents the mean fluorescence intensity value for each nuclei. Data represent the median of the indicated *n* number of nuclei analyzed from three independent experiments. *P*-values were obtained using ordinary one-way ANOVA. **D** Immunoblot of PAR in cell lines treated with 10 μM PARGi and with 20 min DL. α-tubulin used as a loading control. **E** Quantitation of PAR in **D**, normalized to α-tubulin. Data are mean ± SD from three independent experiments; *P*-values were obtained using ordinary one-way ANOVA. **F** Percent β-galactosidase positive cells treated with 20 min DL with or without 0.5 μM Olaparib (PARPi). Data are mean ± SD from three independent experiments; *P*-values were obtained using two-way ANOVA. For all panels only comparisons yielding significant *p*-values are shown, except for non-significant comparisons shown in **C** for clarity. Source data are provided as a Source Data file.

and OGG1 cells. Additionally, others have proposed that MUTYH may function in DDR signaling of oxidative damage independently of its excision activity[31]. MUTYH may activate p53 in the absence of OGG1, by impairing replication not only through SSB generation at 8oxoG:A, but also by binding non-productively to 8oxoG:C. Consistent with a tumor suppressor role by preventing mutagenesis, treatments with oxidant KBrO$_3$ increase G to T mutations in both Mutyh KO and Ogg1 KO mice, but only the Mutyh KO mice show increased tumorigenesis[76,77]. We find MUTYH loss in cells lacking OGG1 allows continued growth after chronic telomere damage, despite increased micronuclei-like species, which we propose arise from chromosome breaks since they co-localize with 53BP1, and chromatin bridges also increase[78]. Consistent with increased genomic instability, *Mutyh$^{-/-}$ Ogg1$^{-/-}$* mice are highly predisposed to tumorigenesis[79]. Our results suggest that MUTYH loss combined with oxidative stress promotes cellular transformation not only from an accumulation of mutations, but also from telomere-mediated senescence evasion and increased genomic instability over time.

Evidence indicates that DNA-damage-induced senescence promotes inflammation and related diseases[80]. Therefore, given that senescence is associated with pro-inflammatory responses, our findings have important implications for the role of OGG1 inhibitors in suppressing inflammation by impairing transcription of proinflammatory cytokines in TNFα-challenged cells and mice[39]. OGG1 inhibitors also mitigate bleomycin-induced pulmonary fibrosis and bacterial lung infections in mice, by reducing inflammatory responses[81,82]. Our data suggest that OGG1 inhibitors may also suppress inflammation by preventing senescence and pro-inflammatory SASP caused by oxidative stress at telomeres. However, our finding that OGG1 loss greatly exacerbates senescence caused by chronic telomeric 8oxoG damage, indicates that anti-inflammatory therapies based on OGG1 inhibition could be unsuitable for chronic inflammatory conditions. In summary, our study demonstrates that repair intermediates arising at telomeres from OGG1 and MUTYH glycosylase activity impair telomere replication and activate PARylation and DDR at telomeres, contributing to 8oxoG-mediated cellular senescence. Our data also reveal a potential role for MUTYH in p53-mediated DDR signaling from persistent unrepaired 8oxoG lesions to protect against genomic instability and tumorigenesis.

## Methods
### Cell culture and cell lines generation
BJ hTERT FAP-mCER-TRF1 wild type (BJ FAP-TRF1 WT) were previously described[19] and were grown in DMEM (Gibco) with 10% Hyclone FBS, 1% penicillin/streptomycin and 500 μg/ml G418 (to maintain FAP-TRF1 expression). To generate knockout cell lines, Hek293T cells (ATCC) were transfected with pLentiCRISPR V2 vectors (GenScript) expressing S. pyogenes Cas9 and guide RNAs designed and validated for uniquely targeting the human OGG1 and MUTYH genes[83], and with Mission Packaging Mix (Sigma) to produce lentivirus. Recipient cells were infected with virus collected 48 and 72 h post transfection and then selected as described below. OGG1 knock out (KO) and MUTYH KO cells were obtained by infection of BJ FAP-TRF1 WT with lentivirus expressing respectively guide RNAs targeting *OGG1* exon 4 (gRNA 3:

GCTACGAGAGTCCTCATATG) and *MUTYH* exon 2 (gRNA 5: GCATGC-TAAGAACAACAGTC) and selected with 1.5 μg/ml Puromycin (Gibco). OGG1 KO/MUTYH KO (DKO) cells were obtained by infection of BJ FAP-TRF1 OGG1 KO cells with lentivirus expressing same MUTYH targeting guide RNA as described above, and were selected with 10 μg/ml Blasticidin S HCl (Gibco). After selection and death of uninfected cells, the infected cells were expanded and tested for mycoplasma, and expression of targeted protein(s) was determined by western blotting. These cultures represent bulk populations and are not clonal expansions. Except for Hek293T cells, all cells are maintained at 5% O$_2$.

### Cell treatments
To generate singlet oxygen at telomeres, cells were treated with dye and light (DL). Briefly, cells were plated at an appropriate density for the experiment overnight. The next day, cells were incubated in OptiMEM (Gibco) at 37 °C for 15 min before adding 100 nM MG2I for another 15 min. Cells were then exposed in the lightbox to a high-intensity 660 nm LED light at 100 mW/cm$^2$ for 20 min (unless indicated otherwise) to trigger excitation of the FAP-bound MG2I dye and the production of singlet oxygen. KBrO$_3$ (Sigma) was added in OptiMEM at the indicated concentrations for 1 h. For chronic exposure to telomeric singlet oxygen, cells were treated with 20 min DL as described above, for three consecutive days and harvested/reseeded every fourth day for a total of 24 days and 18 exposures. PDD00017272 (PARGi) (Tocris) was added in OptiMEM at the indicated concentrations for the entire time of incubation with OptiMEM and DL treatment. AZD2281 (olaparib) (Selleck chemicals) was added in complete media at the indicated concentrations after DL treatment and cells were incubated for 4 days.

### Growth analyses
For cell counting experiments, 25,000 cells were plated in 6-well overnight. Cells were treated as indicated and returned to the incubator and recovered for 4 days. Cells were detached from the plates, resuspended, and counted on a Beckman Coulter Counter. Each experimental condition had 3 technical replicates, which were averaged.

### Senescence associated Beta-Galactosidase assay
Detection of β-gal activity was performed using the Senescence β-Galactosidase Staining Kit according to the manufacturer's instructions (Cell Signaling). Briefly, cells were washed with PBS, and then fixed at room temperature for 10 minutes. Cells were washed again twice with PBS and then were incubated overnight at 37 °C with X-gal staining solution with no CO$_2$. Images were acquired with a Nikon brightfield microscope with DS-Fi3 camera and analyzed in NIS Elements. At least 300–800 cells were counted per condition for each experiment.

### Immunofluorescence and FISH
Cells were seeded on coverslips and treated as indicated. Following treatment and/or recovery, cells were washed twice with PBS and fixed at room temperature with 4% formaldehyde (PFA) for 10 min, except for PAR (10H) staining fixation was on ice in -20 °C methanol/acetone

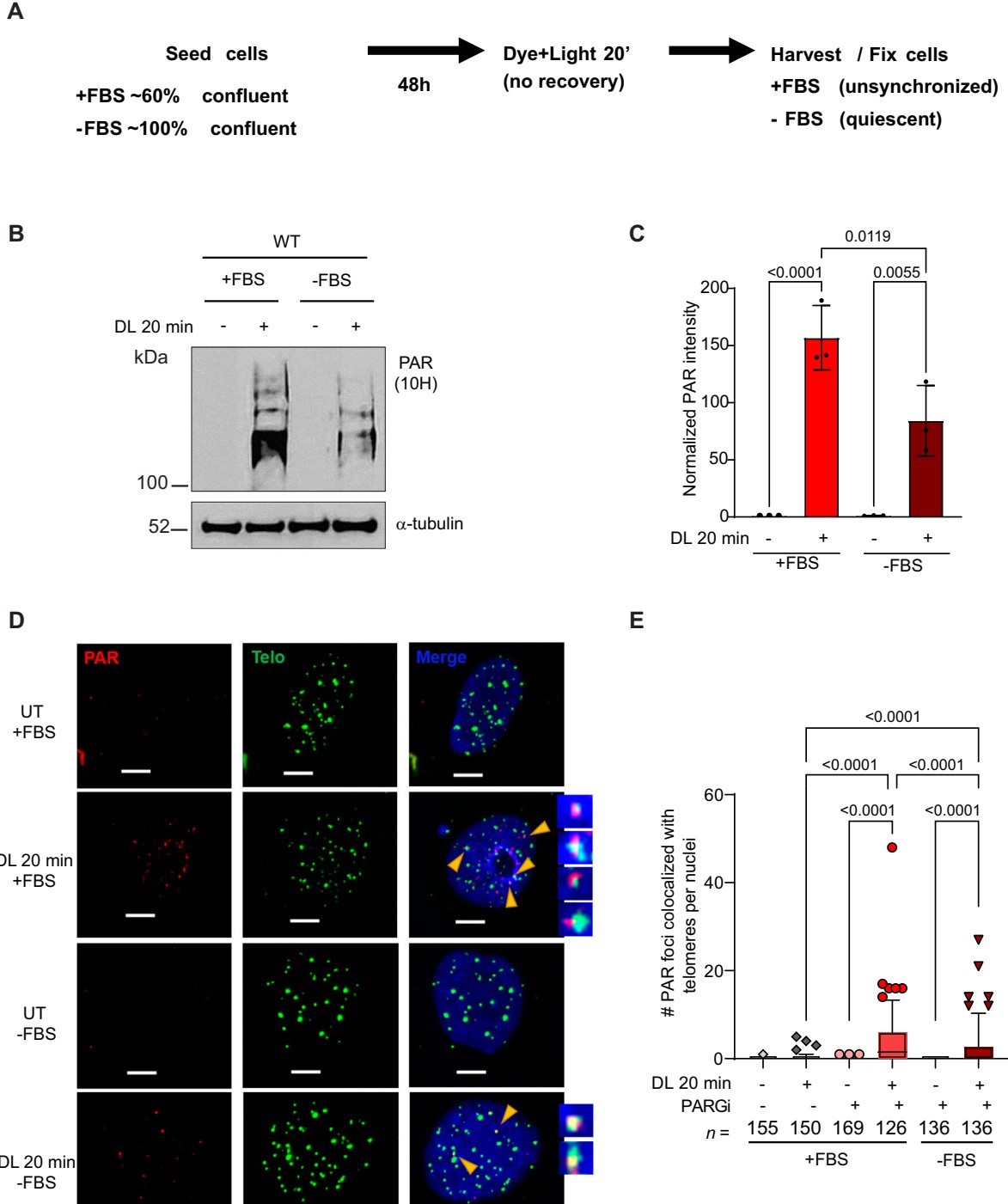

**Fig. 5 | Both BER and replication stress activate PARylation after 8oxoG damage. A** Schematic of experiments for PAR detection by WB (**B**) and telomere IF-FISH (**D**) in unsynchronized (cells grown with 10% FBS (+FBS)) and quiescent (cells grown with 0.1% FBS (-FBS)) WT cells. **B** Immunoblot of PAR in unsynchronized (+FBS) or quiescent (-FBS) WT cells treated with 10 μM PARGi and with no treatment or after 20 min DL. α-tubulin was used as a loading control. **C** Quantitation of PAR in **B**, normalized on α-tubulin. Data are mean ± SD from three independent experiments; *P*-values were obtained using ordinary one-way ANOVA. **D** Representative IF-FISH images showing PAR (red) colocalizing with telomeres (green) by telo-FISH in + or -FBS WT cells with no treatment or after 20 min DL. Yellow arrowheads point to the colocalization of PAR with telomeres. Scale bars = 10 μm. **E** Quantification of the number of PAR foci colocalizing with telomeres per nuclei. Data were obtained from three independent experiments; *n* = total number of nuclei analyzed. In the box-and-whisker plot, the center line represents the median, the box bounds represent the lower and upper quartiles, whiskers indicate the 5–95th percentile range, and dots indicate outliers; *P*-values were obtained using ordinary one-way ANOVA with Tukey's multiple comparisons test. Only comparisons yielding significant *P*-values are shown. Source data are provided as a Source Data file.

(v/v) for 30 min. Fixed cells were rinsed with 1% BSA in PBS, and washed 3x with PBS-Triton 0.2% before blocking with 10% normal goat serum, 1% BSA, and 0.1% Triton-x. Cells were incubated overnight at 4 °C with indicated primary antibodies. The next day cells were washed with PBS-T 3x before incubating with secondary antibodies and washing again 3x with PBS-T. To detect EdU incorporation, both unsynchronized and quiescent cells seeded on coverslips were pulsed with 20 μM EdU after 2 h from 20 min DL treatment and incubated for an

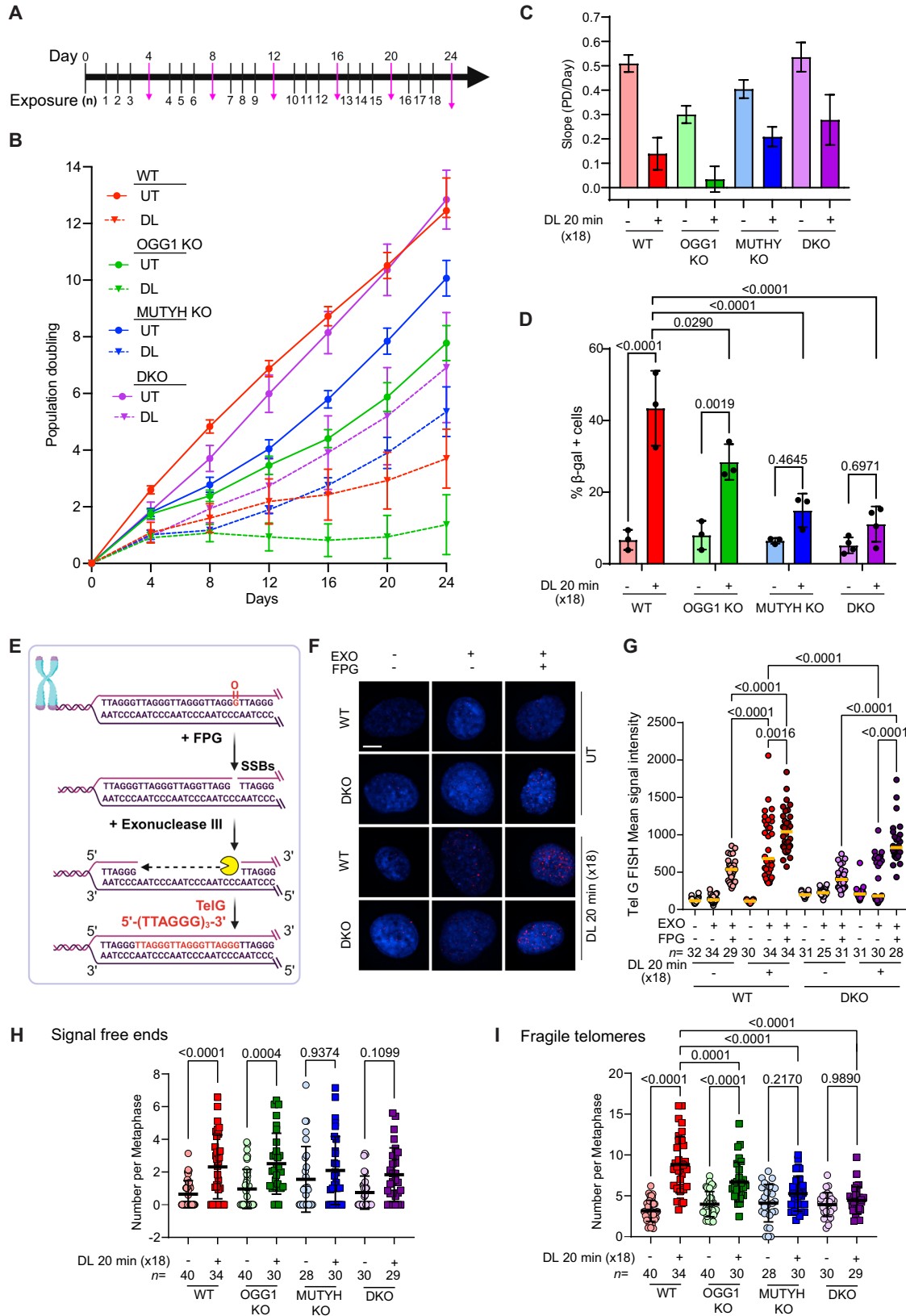

additional hour before being fixed and then Click chemistry was performed according to the manufacturer's instructions (Thermo). When performing FISH, cells were re-fixed with 4% formaldehyde and rinsed with 1% BSA in PBS, and then dehydrated with 70%, 90%, and 100% ethanol for 5 min. The Telomeric PNA probe was diluted 1:100 (PNA Bio, F1004, TelC-Alexa 488, 3xCCCTAA) in hybridization buffer (70%

formamide, 10 mM Tris HCl pH 7.5, 1x Maleic Acid buffer, 1x $MgCl_2$ buffer) and boiled for 5 min at 85 °C before returning to ice. Coverslips were incubated for 10 min on a hot plate at 75 °C and then at room temperature for 2 h in humid chambers in the dark. After two washes in hybridization wash buffer (70% formamide, 10 mM tris HCl pH 7.5), coverslips were rinsed in water before staining with DAPI and

**Fig. 6 | OGG1 loss sensitizes cells to chronic telomere damage while MUTYH loss promotes resistance. A** Schematic of the repeated telomeric 8oxoG inductions. Purple bars indicate days when cells were harvested and not exposed. **B** Population doubling (PD) over 24 days of untreated cells (solid line) and cells treated with DL 20 min each day except every 4th day of harvest (dotted line). Data are mean ± SD from four independent experiments. **C** Slope of simple linear regression (PD/Day) derived from four independent experiments shown in **B**. Error bars represent the 95% confidence intervals of the slope estimate; center indicates slope of the regression line. **D** Percent β-galactosidase positive cells exposed 18 times to 20 min DL over 24 days. Data are mean ± SD from three independent experiments; *P*-values were obtained using two-way ANOVA. Comparison of treated and untreated by two-tailed paired t-test only for MUTYH KO or DKO yielded a *p*-value of 0.064 and 0.030, respectively. **E** Schematic for detection of telomeric SSB repair intermediates or of 8oxoG lesions converted to SSBs by FPG pre-incubation using exo-FISH. Created in BioRender. De Rosa (2024) https://BioRender.com/o13d008.

**F** Representative images of exo-FISH performed in cells untreated or treated 18 times with 20 min DL, harvested 30 min after the last treatment, showing SSBs on the telomeric G-rich strand (red foci). Samples were pre-treated with or without FPG and exonuclease III as indicated, then hybridized with TelG FISH probe. Scale bar = 10 μm. **G** Quantification of exo-FISH signal intensity. Each data point represents the mean fluorescence intensity for each cell. Data represent the median of the indicated *n* number of nuclei analyzed per each condition from two independent experiments. Only comparisons yielding significant *P*-values are shown; *P*-values were obtained using two-way ANOVA. Number of telomeric signal-free ends (**H**) and fragile telomeres (**I**) per metaphase in cells transiently depleted for p53 during the last three days of DL exposures. Data represent mean ± SD from the indicated *n* number of metaphases analyzed from two independent experiments; *P*-values obtained using two-way ANOVA comparing all the cell lines. Source data are provided as a Source Data file.

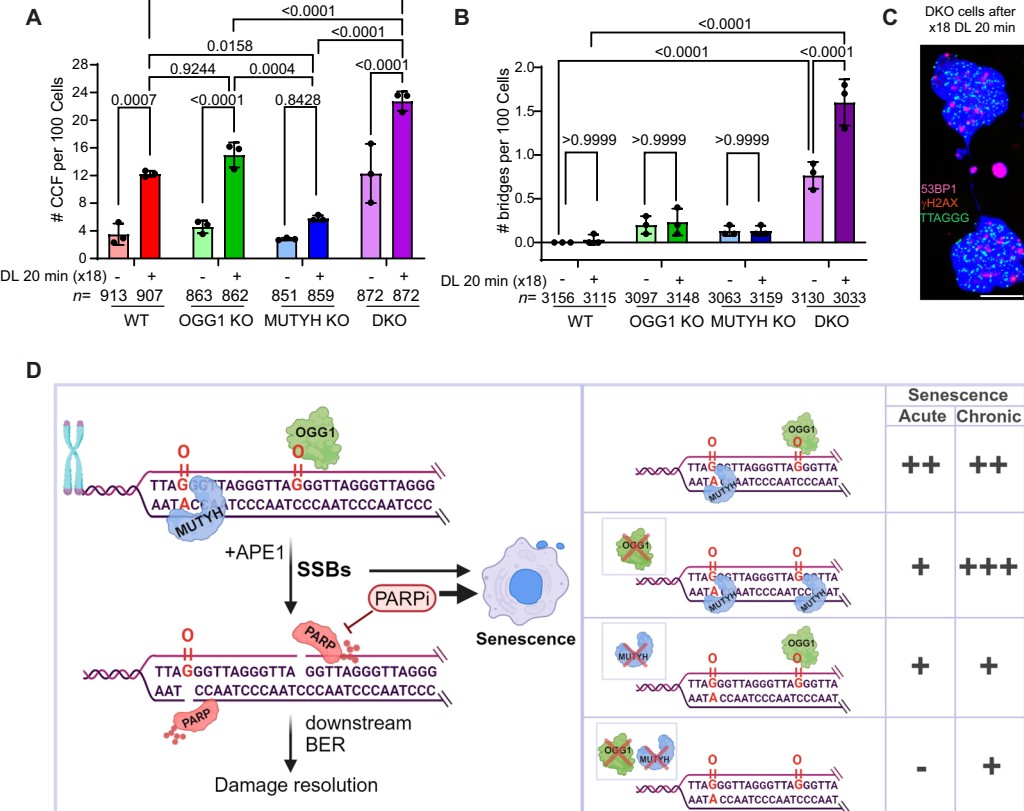

**Fig. 7 | Chronic telomere damage drives chromosomal instability in OGG1 and MUTYH doubly deficient cells. A** Quantification of cytoplasmic DNA species (CCF) foci 24 h after recovery from last exposure to 20 min DL (x18). Data are mean ± SD from three independent experiments; *n* = total number of cells analyzed. *P*-values were obtained using two-way ANOVA. **B** Quantification of the number of bridges per 100 cells counted. Data are mean ± SD from three independent experiments; *n* = total number of cells analyzed. *P*-values were obtained using two-way ANOVA. **C** Representative image of cytoplasmic DNA species and chromatin bridge in DKO cells after 24 h from the last of 18 exposures. Scale bar = 10 μm. **D** Model for how telomeric 8oxoG processing drives cellular senescence after acute and chronic damage. (Left) In WT cells the activity of both BER glycosylases OGG1 and MUTYH at telomeric 8oxoG lesions and 8oxoG:A mismatches, increases SSB repair intermediates, that drive senescence after acute and chronic telomeric 8oxoG

formation. PARP inhibition (PARPi) exacerbates senescence induction after telomere damage by retaining PARP1/2 at SSBs sites. (Right) In OGG1 KO cells, MUTYH-generated SSBs promote senescence after acute damage, while futile BER cycles and possible interaction with unrepaired 8oxoGs provoke extensive senescence after chronic damage. In MUTYH KO cells, OGG1-generated SSBs promote senescence after acute and chronic damage. Lack of glycosylase activity in DKO cells prevents telomeric 8oxoG-induced SSBs, suppresses senescence after acute damage, and attenuates growth reduction after chronic damage despite an accumulation of unrepaired 8oxoG lesions. The degree of senescence induction is indicated by the number of + signs for comparison, while the - sign indicates no senescence or growth reduction. Created in BioRender. De Rosa, M. (2024) https://BioRender.com/y04p503. Source data are provided as a Source Data file.

mounting with ProLong™ Diamond Antifade Mountant (Thermo Fisher). For EdU+ cell counting, cells were pulsed with 20 μM EdU (Thermo) for 1 hour before being fixed in PFA, permeabilized with PBS-Triton 0.2% and blocked as described above. Cells were then treated according to the manufacturer instructions of the Click-iT™ EdU Cell

Proliferation Kit for Imaging, Alexa Fluor™ 488 dye (Thermo Fisher). Click chemistry with Alexa Flour 488 azide was carried out for 30 minutes in the dark, before coverslips were rinsed with water, stained with DAPI and mounted. Image acquisition was performed with a Nikon Ti inverted fluorescence microscope. Z stacks of 0.2 μm (60x

objective) or 0.5 µm (20x objective) thickness were captured and images were deconvolved using NIS Elements Advance Research software algorithm. For MN counting at least 400 nuclei were scored per experiment. For EdU+ cell counting at least 250 nuclei were scored per condition.

Antibodies used were anti-gammaH2AX (Ser139) mouse monoclonal (Santa Cruz #sc-517348, IF dilution 1:200), anti-53BP1 rabbit polyclonal (Novus #NB100-304, IF dilution 1:1000), anti-cGAS rabbit monoclonal (Cell Signaling #15102, IF dilution 1:1000), anti-Poly(ADP-Ribose)10H mouse monoclonal (Enzo Life Science #ALX-804-220 R100, IF dilution 1:1000).

### Analysis of secreted proteins

Cells were treated as indicated, and recovered for 7 days. Medium was collected and debris pelleted by centrifugation for 10 min. Media were stored at −80 °C until ready for analysis. The indicated analytes were assessed for concentration with multiplex ELISA (Luminex). A blank medium sample was analyzed for background levels. For one of the biological replicates, each sample was analyzed in technical duplicate, to assess technical variability of the analysis. After determining concentrations alongside standard curves, the values were adjusted for the number of cells present at the time of harvest.

### Telomere FISH on metaphase spreads

Chromosome spreads were prepared by incubating cells with 0.05 µg/ml colcemid for 2 h prior to harvesting with trypsin. Cells were incubated with 75 mM KCl for 13 minutes at 37 °C and fixed in methanol and glacial acetic acid (3:1). Cells were dropped on washed slides and dried overnight before fixation in 4% formaldehyde. Slides were treated with RNaseA and Pepsin at 37 °C, and then dehydrated. FISH was performed as described above, and included a Pan-centromere probe (PNA Bio, F3005, CENPB-Cy5, ATTCGTTGGAAACGGGA) in addition to the TelC probe.

### Exo-FISH

Exo-FISH was performed as previously described[52,84] with slight modification. Cells were harvested by trypsinization, washed in PBS and 200 K cells were swollen in 0.56% KCl for 20 minutes. After 5 min centrifugation at 300xg, cell pellets were fixed in 3:1 methanol:acetic acid for 20 minutes, and then a consistent number (20 K) was homogeneously spread onto glass slides. After drying overnight at room temperature in the dark, slides were rehydrated in PBS for 5 minutes at room temperature. The following treatments were all performed in a humidified chamber: slides were treated with 0.5 mg/ml RNaseA (Invitrogen) for 10 minutes at 37 °C, and then incubated with buffer only (-exo) or 500 mU/µl exonuclease III (Promega) (+exo) for 30 minutes at 37°. Slides were then serially dehydrated with 70%, 95%, and 100% EtOH, 5 minutes each, and air-dried overnight in the dark.

FISH probe hybridization was then performed onto the dried slides with 200 nM TelG-Cy3 (PNABio) diluted in hybridization solution (10 mM Tris-HCl pH 7.2, 70% formamide, 0.5% blocking solution), [blocking solution: 10% blocking reagent (w/v, Roche) dissolved in maleic acid buffer (100 mM maleic acid, 150 mM NaCl, pH 7.5)] for 90 minutes at room temperature in a humidified chamber. Slides were washed for 15 minutes in hybridization wash buffer #1 (10 mM Tris-HCl pH 7.2, 70% formamide, 0.1% BSA), followed by three washes, 5 minutes each, in hybridization wash buffer #2 (0.1 M Tris-HCl pH 7.2, 0.15 M NaCl, 0.08% Tween-20). After last serial dehydration in 70%, 95%, and 100% EtOH, 5 minutes each, slides were air-dried before mounting with ProLong mounting media containing DAPI (Thermo-Fisher Scientific). For chronic experiments where also FPG was used, cells were harvested and fixed 30 minutes after the last exposure to DL 20 min (N18). All incubations and treatments were performed as described above, except an additional step was added between RNase A and Exonuclease III. Briefly, after RNase A treatment, slides were

washed in PBS three times, 5 minutes each, and then incubated with 4U FPG (in FPG buffer complemented with 100 µg/ml BSA) for 1 hour at room temperature in a humidified chamber. Slides were then washed again as above before proceeding to treatment with exonuclease III.

### Image acquisition and analysis

All IF images were acquired on a Nikon Ti inverted fluorescent microscope equipped with an Orca Fusion cMOS camera, or CoolSNAP HQ2 CCD. Z-stacks were acquired for each image and deconvolved using blind, iterative methods with NIS Elements AR software.

For co-localizations, deconvolved images were converted to Max-IPs and converted to a new document. The object counts feature in NIS AR was used to set a threshold for foci that was kept throughout the experiment. The binary function was used to determine the intersections of 2 or 3 channels in defined regions of interest (ROI) (DAPI stained nuclei). For exo-FISH acquisition and analysis, to prevent signal intensity dependency on cell aggregation and/or uneven spread onto slides, images were taken throughout the slide avoiding imaging aggregates. Exposure time (ms) for both DAPI and Cy3 were kept consistent across experimental conditions in each separate experiment. Background signal was subtracted through blind iterative deconvolution (NIS Elements AR software). After conversion to Max-IPs, ROIs were defined through the DAPI channel, while a threshold of the Cy3 channel was set in order to select relevant foci. Both lookup tables (LUTs) and the threshold were kept constant across experimental conditions throughout the analysis of an independent experiment. The automated measurement of the mean signal intensity for the thresholded channel (Cy3) was used on ROIs.

### siRNA transfections

To deplete endogenous p53 for telomere FISH on metaphase spreads experiments, a single siRNA targeting the human TP53 gene was synthesized and purchased from Ambion (Life Technologies). Briefly, 350,000 cells were seeded per well of a 10-cm dish containing growth medium without antibiotics. The following day cells were transfected with Lipofectamine™ RNAiMAX Transfection Reagent (Thermo Fisher). siRNAs and RNAiMAX were diluted in OptiMEM. A working siRNA concentration of 12.5 nM and 6 µl RNAiMAX transfection reagent per 10-cm plate was used. Transfection media was replaced with complete culture media 24 h later. Cells were treated with 20 min DL after 24 h and collected after the additional 24 h. For the siRNA knockdown experiments, Dharmacon ON-TARGETplus SMARTpool siRNAs for human OGG1 (#L-005147-03-0005), MUTYH (#L-012806-00-0005) and APE1 (#L-010237-00-0005) were used with the same protocol as above, with the exception that 30,000 cells were seeded per well of a 35-mm dish and a working siRNA concentration of 5 nM and 3 µl RNAiMAX transfection reagent was used. Knockdown efficiency was validated by Western blot.

### Western blotting

Cells were collected from plates with trypsin, washed with PBS, and then lysed on ice with RIPA buffer (Santa Cruz) supplemented with PMSF (1 nM), 1x Roche Protease and Phosphatase Inhibitors, and Benzonase (Sigma E8263; 1:500) for 15 min and then incubated at 37 °C for 10 min, before spinning down at 15,000 rpm for 10 minutes at 4 °C. Protein concentrations were determined with the BCA assay (Pierce) and 20-30 µg of protein was electrophoresed on 4–12% (or 12% for OGG1 blot) Bis-Tris gels (Thermo) before transferring to PVDF or nitrocellulose membranes (GE Healthcare). For PAR 10H western blots, cells were directly lysed right after treatment in 4× LDS sample buffer completed with 20 µM PARPi and 20 µM PARGi. Proteins were denatured for 5 min at 95 °C to inactivate PARP and PARG enzymes. Samples were then gently homogenized using universal nuclease (Pierce/ThermoFisher), resolved by SDS−PAGE electrophoresis, and transferred onto nitrocellulose. Red Ponceau staining was

performed to ensure even transfer of proteins onto membranes, which were then washed in TBS-T and blocked in 5% milk, or 5% BSA for phosphorylated proteins, and blotted with primary and secondary HRP antibodies. The signal was detected by ECL detection and X-ray film.

Antibodies used were anti-MUTYH mouse monoclonal (Abnova #H00004595-M01, WB dilution 1:500), anti-OGG1 rabbit monoclonal (Abcam #ab124741, WB dilution 1:1000), anti-alpha-tubulin mouse monoclonal (Millipore #05-829, WB dilution 1:5000), anti-TRF1 (TRF-78) mouse monoclonal (Santa Cruz #sc-56807, WB dilution 1:1000), Anti-beta-actin mouse monoclonal (Cell Signaling #3700, WB dilution 1:5000), anti-Phospho-Chk1 (ser345) rabbit monoclonal (Cell Signaling #2348, WB dilution 1:500), anti-Chk1 (2G1D5) mouse monoclonal (Cell Signaling #2360, WB dilution 1:500), anti-GAPDH mouse monoclonal (Santa Cruz #sc-47724, WB dilution 1:5000), anti-Phospho-ATM (ser1981) rabbit monoclonal (Abcam #ab81292, WB dilution 1:2000), anti-ATM mouse monoclonal (Sigma #A1106, WB dilution 1:500), anti-Phospho-Chk2 (Thr68) rabbit monoclonal (Cell Signaling #2197, WB dilution 1:500), anti-Chk2 mouse monoclonal (Cell Signaling #3440, WB dilution 1:500), anti-p21 rabbit monoclonal (Cell Signaling #2947, WB dilution 1:3000), anti-p53 mouse monoclonal (Santa Cruz #sc-126, WB dilution 1:200), anti-Poly(ADP-ribose) (10H) mouse monoclonal (Enzo Life Science #ALX-804-220-R100, WB dilution 1:500), anti-APE1 rabbit monoclonal (Cell Signaling #10519S, WB dilution 1:1000), anti-Cyclin A mouse monoclonal (Santa Cruz #sc-271682, WB dilution 1:200).

### Flow Cytometry

Analysis of apoptosis was performed using the Dead Cell Apoptosis Kit according to the manufacturer's instructions (Thermo Fisher). Cells were treated as indicated, and allowed to recover for 4 days. Floating cells were collected, and then attached cells were collected with trypsin and combined. After spinning down and washing, cells were incubated with Alexa Fluor 488 annexin V and $1\,\mu g/ml$ propidium iodide in 1x annexin-binding buffer for 15 min in the dark (Thermo). After resuspending in the additional binding buffer, cells were analyzed on a CytoFLEX S Flow Cytometer (Beckman) using FL1 and FL3. Preliminary standard gating for cells versus debris and singlet, and analysis of the results, were conducted with FlowJo™ v10.8 Software (BD Life Sciences).

### Synthesis of OGG1i (TH5487) and OGG1i[NA]

4-(4-Bromo-2-oxo-2,3-dihydro-1*H*-benzo[*d*]imidazol-1-yl)-*N*-(4-iodo-phenyl)piperidine-1-carboxamide (TH5487 (OGG1i)). This compound was prepared according to literature data[39] using the following procedures. A mixture of 4-aminopiperidine-1-carboxylic acid *tert*-butyl ester (1.5 g, 7.5 mmol), 1-bromo-3-fluoro-2-nitro- benzene (1.5 g, 6.8 mmol), and diisopropylethylamine (1.7 mL, 10.2 mmol) was stirred in a sealed vial at 120 °C for 16 h. The mixture was concentrated and purified by chromatography on $SiO_2$ (50–100% $CH_2Cl_2$ in hexanes) to afford *tert*-butyl 4-((3-bromo-2-nitrophenyl)amino)piperidine-1-car-boxylate (2.27 g, 83%) as an orange colored foam: IR (ATR, neat) $\nu_{max}$ 3407, 2976, 2932, 2864, 1684, 1598, 1563, 1532, 1497, 1478, 1451, 1424, 1365, 1276, 1239, 1167, 1140, 1094, 1056 cm$^{-1}$; $^1$H NMR (600 MHz, CDCl$_3$) δ 7.14 (t, $J = 8.1$ Hz, 1 H), 6.94 (d, $J = 7.8$ Hz, 1 H), 6.76 (d, $J = 8.4$ Hz, 1 H), 5.60 (d, $J = 6.6$ Hz, 1 H), 4.00 (bs, 2 H), 3.54-3.50 (m, 1 H), 2.98 (m, 2 H), 2.00 (d, $J = 10.8$ Hz, 2 H), 1.46 (s, 9 H), 1.44-1.40 (m, 2 H); $^{13}$C NMR (150 MHz, CDCl$_3$) δ 154.7, 142.3, 136.9, 133.0, 121.9, 116.6, 112.8, 80.0, 50.0, 31.8, 28.5; HRMS (ESI$^+$) *m/z* for C$_{16}$H$_{23}$O$_4$N$_3$Br, [M + H]$^+$: calcd 400.0867, found 400.0871.

To a stirred solution of *tert*-butyl 4-((3-bromo-2-nitrophenyl)amino) piperidine-1-carboxylate (2.25 g, 5.62 mmol) in acetonitrile (24 mL) and water (2.4 mL) was added NiCl$_2$ (0.146 g. 1.12 mmol) at room temperature. The reaction mixture was cooled to 0 °C and NaBH$_4$ (0.85 g, 22.5 mmol) was added portionwise over ca. 10 min. The mixture was diluted with CH$_2$Cl$_2$ (30 mL), decanted, poured into NaHCO$_3$ (30 mL), and extracted with CH$_2$Cl$_2$ (3 × 40 mL). The combined organic layers

were dried (Na$_2$SO$_4$), filtered, and concentrated. The crude residue was purified by chromatography on SiO$_2$ (10%–20% EtOAc in hexanes) to afford *tert*-butyl 4-(2-amino-3-bromoanilino)piperidine-1-carboxylate (1.94 g, 93%) as a yellow oily foam: IR (ATR, neat) $\nu_{max}$ 3389, 2975, 2932, 2856, 1672, 1585, 1452, 1426, 1392, 1366, 1344, 1311, 1275, 1239, 1218, 1167, 1138, 1069 cm$^{-1}$; $^1$H NMR (400 MHz, CDCl$_3$) δ 6.93 (dd, $J = 1.6, 7.6$ Hz, 1 H), 6.63 (t, $J = 7.8$ Hz, 1 H), 6.59 (dd, $J = 1.6, 8.0$ Hz, 1 H), 4.02 (bs, 2 H), 3.79 (bs, 1 H), 3.39-3.32 (m, 2 H), 2.92 (app t, $J = 11.8$ Hz, 2 H), 2.01-1.97 (m, 2 H), 1.46 (s, 9 H), 1.41-1.30 (m, 2 H); $^{13}$C NMR (100 MHz, CDCl$_3$) δ 154.8, 136.6, 133.3, 122.4, 120.5, 112.2, 111.5, 79.6, 50.3, 32.3, 28.5; HRMS (ESI$^+$) *m/z* for C$_{16}$H$_{23}$O$_4$N$_3$Br, [M + H]$^+$: calcd 370.1125, found 370.1125.

A mixture of *tert*-butyl 4-(2-amino-3-bromoanilino)piperidine-1-carboxylate (1.94 g, 5.24 mmol), and diisopropylethylamine (2.6 mL, 15.7 mmol) was stirred in CH$_2$Cl$_2$ (10 mL) at 0 °C. A solution of tri-chloromethyl carbonochloridate (0.32 mL, 2.6 mmol) in CH$_2$Cl$_2$ (30 ml) was added dropwise. After 30 min, the mixture was concentrated and purified by chromatography on SiO$_2$ (0%–6% MeOH in CH$_2$Cl$_2$) to give *tert*-butyl 4-(4-bromo-2-oxo-2,3-dihydro-1*H*-benzo[*d*]imidazol-1-yl) piperidine-1-carboxylate (1.95 g, 94%) as a slightly brown foam: IR (ATR, neat) $\nu_{max}$ 3158, 2975, 2932, 2864, 1693, 1622, 1593, 1487, 1461, 1424, 1365, 1321, 1287, 1274, 1243, 11655, 1118 cm$^{-1}$; $^1$H NMR (400 MHz, CDCl$_3$) δ 10.39 (s, 1 H), 7.15 (dd, $J = 0.6, 8.0$ Hz, 1 H), 7.04 (d, $J = 8.0$ Hz, 1 H), 6.90 (t, $J = 8.0$ Hz, 1 H), 4.50-4.42 (m, 1 H), 4.30 (bs, 1 H), 2.90-2.78 (m, 2 H), 2.32 (d, $J = 4.4, 8.8$ Hz, 1 H), 2.28 (d, $J = 4.4, 8.8$ Hz, 1 H), 1.81 (dd, $J = 10.4$ Hz, 2 H), 1.48 (s, 9 H); $^{13}$C NMR (100 MHz, CDCl$_3$) δ 154.7, 154.5, 129.7, 128.0, 124.0, 122.2, 108.2, 102.8, 80.0, 51.1, 29.2, 28.5; HRMS (ESI$^+$) *m/z* for C$_{16}$H$_{23}$O$_4$N$_3$Br, [M + H]$^+$: calcd 396.0917, found 396.0913.

To a stirred solution of *tert*-butyl 4-(4-bromo-2-oxo-2,3-dihydro-1*H*-benzo[*d*]imidazol-1-yl)piperidine-1-carboxylate (1.95 g, 4.92 mmol) in CH$_2$Cl$_2$ (150 mL) was added TFA (12 mL) at 0 °C. The reaction mixture was warmed to room temperature, stirred for 2 h, concentrated under reduced pressure, and dried *in vacuo*. The crude residue was used in the next step without further purification.

A solution of crude residue (0.285 g) from the previous step in CH$_2$Cl$_2$ (10 mL) was treated with diisopropylethylamine (0.17 mL, 1.0 mmol) and stirred in a sealed tube at room temperature for 5 min. A solution of 4-iodophenylisocyanate (0.14 g, 0.58 mmol) in CH$_2$Cl$_2$ (5 mL) was added and the mixture was heated at 50 °C for 30 min, cooled to room temperature, and stirred for 2 h. The precipitate was collected by filtration and was washed sequentially with CH$_2$Cl$_2$, water, MeOH, and CH$_2$Cl$_2$. The solid was then suspended in boiling MeOH and filtered to give 4-(4-bromo-2-oxo-2,3-dihydro-1*H*-benzo[*d*]imidazol-1-yl)-*N*-(4-iodo-phenyl)piperidine-1-carboxamide (0.22 g, 80%) as a cream-white solid: IR (ATR, neat) $\nu_{max}$ 3261, 2847, 1707, 1638, 1584, 1516, 1485, 1459, 1424, 1372, 1324, 1311, 1300, 1280, 1242, 1157, 1109 cm$^{-1}$; $^1$H NMR (400 MHz, DMSO-d$_6$) δ 11.31 (s, 1 H), 8.69 (s, 1 H), 7.57-7.55 (m, 2 H), 7.37-7.35 (m, 2 H), 7.25 (d, $J = 7.8$ Hz, 1 H), 7.16 (d, $J = 7.8$ Hz, 1 H), 6.95 (t, $J = 8.1$ Hz, 1 H), 4.42-4.37 (m, 1 H), 4.28 (d, $J = 13.2$ Hz, 2 H), 2.93 (app t, $J = 12.0$ Hz, 2 H), 2.28 (dd, $J = 4.2, 12.6$ Hz, 1 H), 2.24 (d, $J = 4.2, 12.6$ Hz, 1 H), 1.74 (app dd, $J = 1.8, 10.2$ Hz, 2 H); $^{13}$C NMR (150 MHz, DMSO-d$_6$) δ 154.4, 153.4, 140.7, 136.9, 130.3, 127.8, 123.3, 121.9, 121.7, 107.7, 101.1, 84.6, 50.5, 43.4, 28.6.

*N*-(4-chlorophenyl)-4-(2-oxo-2,3-dihydro-1*H*-benzo[*d*]imidazol-1-yl)piperidine-1-carboxamide (OGG1i[NA]). This compound can be obtained analogously according to the experimental protocol provided for OGG1i. The purity of final products was assessed using an Agilent Technologies 1260 Infinity II LC at 220 nm UV absorption (Waters XBridge BEH C$_{18}$ 2.1 × 50 mm, 2.5 μm) or an Agilent Technologies 385-ELSD (Microsolv Cogent 2.0 Bidentate C$_{18}$ 2.1 × 50 mm, 2.2 μm; ELSD conditions: evaporator and nebulizer set at 45 °C; gas flow set at 1.80 standard liter/min). All final assay samples showed a purity >95% by LCMS analysis with UV (220 nm) and ELS detection.

### Statistics and reproducibility

The number of biological and technical replicates are noted in all Fig. legends and methods. All statistical analysis was performed in

GraphPad Prism 9. No statistical method was used to predetermine sample size. Investigators were not blinded to allocation during experiments and outcome assessments.

### Reporting summary
Further information on research design is available in the Nature Portfolio Reporting Summary linked to this article.

## Data availability
All data generated or analyzed during this study are included in this published article and its supplementary information files. Further information and requests for reagents should be directed to and will be fulfilled by the corresponding author. Source data are provided with this paper.

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

## Acknowledgements

We are grateful to Brigitte Schmidt and Bruce Armitage (Carnegie Mellon University) for providing MG2I dye, and we dedicate this manuscript to the late Marcel Bruchez for his development of the chemoptogenetic tool. We thank Fatimah Adisa and Madalyn Fry for technical assistance with some western blotting experiments. This work was supported by NIH grants K99ES035871 (to M.D.R.), F32AG067710, K99AG033771 (to R.P.B.), R01NS117000 (to P.W.), and R35ES030396 and R01CA207342 and a grant from the Richard King Mellon Foundation (to P.L.O.). This project used the UPMC Hillman Cancer Center Cytometry Facility which is supported in part by award P30CA047904, and the UPMC Cancer Proteomics Facility: Luminex Core Laboratory.

## Author contributions

M.D.R. and P.L.O. conceived the study and designed the experiments. M.D.R. performed most of the experiments. R.P.B. generated the BJ FAP-

TRF1 WT, p53 knock out, and OGG1 knock out cells. A.C.D. assisted with telo-FISH experiments. P.W. and P.R.N. synthesized the active and inactive OGG1 inhibitors. M.D.R. and P.L.O. wrote the manuscript with assistance from the other authors.

## Competing interests

The authors declare no competing interests.
