## [Transparent Peer Review file · Nature Communications]

OGG1 and MUTYH repair activities promote telomeric 8-oxoguanine induced senescence in human fibroblasts

Corresponding Author: Dr Patricia Opresko

Editorial Note: Parts of this Peer Review File have been redacted as indicated to maintain the confidentiality of unpublished data

Version 0:

Reviewer comments:

Reviewer #1

(Remarks to the Author)

In this report, the authors used their chemoptogenetic tool (FAP-TRF1) expressed in BJ hTERT cells to evaluate the impact of oxidative damage to telomeres. Further, they evaluated the response as regulated by the base excision repair proteins OGG1 and MUTYH, two glycosylases that are selective for oxidative DNA damage. OGG1 removes the oxidative base lesion 8oxoG while MUTYH removes the A base opposite the oxidative base lesion 8oxoG. That removal (repair) process initiates base excision repair (BER) that could lead to the accumulation of DNA breaks and elevated PARP1/PARP2 activation if BER is not completed. From these studies they suggest that BER failure of such lesions at telomeres leads to the accumulation of single strand breaks that can initiate senescence and disrupt telomere function.

This is a well-designed and detailed study but some major concerns include:

- 1) The study is limited to a single cell line (human BJ hTERT fibroblasts expressing FAP-TRF1; BJ FAP-TRF1).
- 2) The study is limited to a single KO for OGG1, a single KO for MUTYH and one double KO. It would be expected that a second set of KOs was developed to confirm some of the key findings, and also in at least a second cell line.
- 3) It is noted that the KO cells were not complemented with an expression vector for OGG1 or MUTYH. It would be expected that such complemented cells would be developed to confirm the reversal of some of the key findings.
- 4) The authors show an increase in phospho-CHK1 after DL treatment in Figure 3 and then an increase in phospho-CHK2 in Figure S5. These markers represent different aspects of the DDR and so both should be evaluated in both datasets.
- 5) The almost opposing effects seen when comparing acute to chronic exposure is confounding. Can the authors either textually or experimentally expand on this aspect of the study?
- 6) The proposed model suggests that "chronic telomeric 8oxoG production revealed divergent roles for OGG1 and MUTYH in suppressing or promoting, respectively, damage-induced senescence or apoptosis" as depicted in Fig. 7D. This is confounding as both enzymes initiate BER in response to the 8oxoG base lesion. More explanation and maybe experimental analysis is needed to validate this claim.

Some minor concerns include:

Figure 1E – regarding the stats when comparing the OGG1-KO cells + DL to the DKO cells. + DL, nothing is indicated. Is that meant to suggest not significant?

Figure 1G – regarding the stats when comparing the OGG1-KO cells + DL to the DKO cells. + DL, nothing is indicated. Is that meant to suggest not significant?

Figure 2B – regarding the stats when comparing the OGG1-KO cells + DL to the DKO cells. + DL, nothing is indicated. Is that meant to suggest not significant?

Figure 2F – regarding the stats when comparing the OGG1-KO cells + DL to the DKO cells. + DL, nothing is indicated. Is that meant to suggest not significant?

Figure 2G – regarding the stats when comparing the OGG1-KO cells + DL to the DKO cells. + DL, nothing is indicated. Is that meant to suggest not significant?

Figure 3B – regarding the stats when comparing the OGG1-KO cells + DL to the DKO cells. + DL, nothing is indicated. Is that meant to suggest not significant?

Figure 4F – regarding the stats when comparing the OGG1-KO cells + DL + Olaparib to the DKO cells. + DL + Olaparib, nothing is indicated. Is that meant to suggest not significant?

Reviewer #2

(Remarks to the Author)

In this manuscript the authors describe results that support the conclusion that glycosylase activity on oxidized bases in the telomeres elicits toxic effects such as diminished cell proliferation in the presence of oxidized purines. The concept that repair intermediates can be worse than the offending base is consistent with orthogonal studies of AAG-driven BER imbalance. Importantly, this manuscript takes this paradigm a step further, pointing to the critical role of BER at telomeres as a modulator of cell survival, senescence, and stress responses. The results from these studies have important implications, since telomere stability plays such an important role in modulating phenotypes associated with diseases and aging. A remarkable finding is how powerful DNA damage at telomeres can be at driving biological responses, despite their being relatively small compared to the entire genome. It is noteworthy that the approach of looking at single and double KOs is elegant because the single KOs won't have opposing SSBs, by definition given the sequence at telomeres. The extent to which single strand breaks (without opposing single strand breaks) can trigger complex cellular phenotypes, including a DNA damage response and senescence is an important finding. Another insight is that the double KO OGG1^{-/-}/MUTYH^{-/-} mice are prone to increased mutations (prior work) while at the same time reducing toxicity (this manuscript) which would conspire to increase viable mutant cells, thus driving cancer. The manuscript is expertly written with deep analysis. The combination of research excellence and exciting impact make this manuscript appropriate for publication in Nature Communications.

Overall, the quality of the data is sound and the appropriate techniques have been employed. There are some more minor conclusions that might be adjusted per the recommendations below. The results of this work significantly advance the field as well as helping to explain important biological observations. Below, please find general comments as well as comments that are specific to particular statements/figures.

General: The fact that 8oxoG can only be on one strand is of vital importance to interpreting the results and should be made clear at the outset, possibly in the introduction. This is an important aspect of the overall experimental design that would be obvious to someone who studies telomeres, but less obvious to a more general audience.

General: Of note, OGG1 removes more than just 8oxoG. This should be clarified.

General: A lot is resting on Figure 1B for which there are only three points for each condition. Given that the results for Figure 1B are not consistent with the results in Supp Fig. 1B, it is important that confidence be at its highest for Figure 1B. Specifically, it is not clear why at 5 min of exposure each of the single KOs do not look different from WT while in Figure 1B they do look to be different. For this reason, it is recommended that the results in Figure 1B be repeated such that there are twice the number of values (6 values rather than 3 values for each bar). Of note, in many figures, an additive effect of the DKO is not evident. For example, in Figure 2B the DKO does not look different from the singles. This is also true for Figure 1E. This needs to be addressed.

General: On line 136 it is stated that telomeres are more sensitive than the bulk genome. However, the method for introducing oxidative damage to the telomeres likely leads to clustering of damage that would not necessarily be the case for the global genome. Close clustering can lead to BER intermediates on opposing strands, leading to DSBs. This likely is not the case for cells exposed to KBrO₃. Without knowing total number of lesions or lesions per kb, it is hard to interpret this comparison.

General: The text suggests that cGAS-STING may be activated by CCFs and more so in the WT cells (line 168 in reference to Figure 2F). However, proper analysis of cGAS-STING activity requires more than just looking at pro-inflammatory cytokines, especially since not all of those examined are downstream of STING activation. Furthermore, it is not clear why CXCL1, FAS, GDF-15 and MMP3 are shown in the main figure while GM-CSF, IL-6, IL-7, IL-8, and IL-15 are in the supplement. Of note, IL-6 is downstream of STING and the results show the opposite of the main conclusion (e.g., the DKO has more of an induction of IL-6, not less). This is particularly relevant given the recent results by the Boulton laboratory showing that it is often the case that DNA damage does not induce STING activation. Rather than drawing conclusions about STING activation, perhaps it would be more conservative to focus on which aspects of the results reflect pro-inflammatory responses.

General: Figure 5 could be significantly strengthened. It appears that PAR levels were only assessed in one experiment; quantification across multiple experiments is wanted. Furthermore, the difference between +FBS and -FBS for PARGi+DL is

not immediately apparent by eye. There is a considerable amount of variability, and the magnitude of the effect is slight. It is suggested that either additional experiments be performed, or perhaps to omit this figure. Additionally, while the data in Figure 5D shows the results from three independent experiments, the number of values is greater than 6 for the DL, PARGi - FBS bar. Clarifying how the analysis was done would be helpful.

General: It is noteworthy that there is reduced cell growth for the single knockouts but not so much in the DKO in both Supp Fig. 1F and Figure 6B.

General: The paragraph beginning with line 312 is a bit hard to follow. Confusion stems from the change in the definition of CCF to make it CCF/MN. There is also a statement about MN arising from chromosome breaks (line 324) which introduces a new variable not discussed in other parts of the paper and for which the data have no error bars (Supp Figure 7G). Also, the interpretation would be very different if in Figure 7A the data were plotted as relative increase, in which case the WT look to have a more sensitive response than the DKO, which would be more in sync with the earlier results. Finally, Figure 7D could be improved if the "+" notation was done for both acute and chronic responses.

General: It would be helpful if in the results section there was some explanation for why there was a change in the protocol from a single DL exposure to multiple exposures over time. What is the hypothesis that motivated the long-term exposure paradigm?

Line 54. It is stated that 8oxoG is more likely to form on telomeres. This reviewer is confused as to why/how this is the case and the extent to which there is solid prior work supporting this conclusion.

Line 75. This reviewer would like to know what the relationship is rather than just stating that there is a role.

Line 108: It is stated that the DKO is rescued more than the single KOs. The issue at hand is whether or not 73% relative cell number for OGG1 KO is significantly different from 85% for the DKO. Statistics should be performed before this conclusion can be stated. Certainly one can say there is a trend.

Line 123: In reference to Figure 1C, it is stated that OGG1 is different from WT. While statistically 43% is less than 51%, this is a very slight difference and likely would not hold up if the entire experiment were repeated three times more. As such, it should be made clear that while the results are statistically significant, the magnitude of the difference is slight.

Figure 2B: It would be helpful if something was said about the fact that the DKO is not different from the singles.

Line 159: Are bridges the only way to get CCF? If not, then this statement needs to be adjusted to indicate that other mechanisms of CCF production had not been explored and could help to explain why the glycosylase KO cells have more CCF.

Supp Figure 3: It is interesting that the single knockouts do not show attenuation of pATM or pChk2 while the DKO does. This is inconsistent with Figure 1E. This is curious and warrants mention at least, if not discussion.

Figure 4C: The MUTYH results are somewhat confusing. While the suggestion that MUTYH might assist OGG1 is interesting, much of the manuscript would need to be adjusted in terms of interpretation if this is an important mechanism of action. Interestingly, if this is a dominant activity of MUTYH, that would help to explain why in so many of the results one does not see an additive effect in the DKO. This requires deeper thought and possible adjustment of conclusions in other parts of the manuscript.

Figure 6B: Another way to look at the data is to plot the final values at 24 days. A rough estimate is: WT 25%, OGG1 14%, MUTYH 50%, DKO 54%.

Line 285. This reviewer doesn't see a 'plateau'. Perhaps omit this descriptor.

Figure 6G: There looks to be many more points for the WT than for the DKO. Since the spread will increase with increasing number of values, these should be made equal.

Figure 7A: Is this CCF per micronucleus? The axis appears different from Figure 2, which is a bit confusing. Perhaps the analysis of CCF metrics should be consistent between Figure 2 and Figure 7.

Supp Figure 7C. The results for the DKO look oddly absolute. If this were an analysis of phospho-P53 (which is the preferred approach), this might make sense. However, the results here suggest that there is no p53 at all in the untreated cells, which is unexpected. It is politely suggested that this experiment be repeated.

Line 350: It is suggested that the results for the chronic exposure be discussed in a separate paragraph from the main results since the findings are not as solid and are at times contradictory. This separate paragraph can also lay out the hypothesis, as mentioned above.

Paragraph starting on line 355. Many references are missing. Several laboratories have done work on imbalanced BER. References are needed for the first and second sentences.

Line 373: It is odd that this paragraph implies that MPG is not the same enzyme as AAG, which it is.

Line 391: It is not clear from the results that the double KO is significantly different from OGG1 in Figure 1B, which would mean softening this statement in the discussion.

Line 400: If there are not published data showing that unproductive binding of MUTYH causes PARylation, this statement should be removed.

Line 405: It is noteworthy that a lot of the biology happens in the absence of replication. There is still PARylation in serum starved cells (Figure 5B) and the results in Figure 5D are not entirely convincing. Perhaps temper the conclusion here or do additional experiments.

Line 407: If plotted as relative to untreated, the results are quite different. This is how the data were presented in Figure 1B, which is a centrally important figure.

Line 422 and Figure 7D: What is the definition of "Genomic Instability"?

Line 432: It is important to put the results of this manuscript into the context of what is known via the studies by Istvan Boldogh.

Line 443: This doesn't seem to be the strongest final statement. There is not really great evidence that MUTYH is acting on persistent unrepaired 8oxoG since the results can be explained by repeated induction of 8oxoG, which is different from the concept that a lesion is sitting there unrepaired for a significant duration. It would be helpful if the authors made it a bit more clear what are the major/novel impacts of this work.

Details:

Line 63. Remained.

Line 108: Define DL.

Line 116: reverse the order in the parenthesis to read Supp figure number before the main figure number.

Reviewer #3

(Remarks to the Author)

Telomeres are critical for maintaining chromosome stability but are susceptible to oxidative damage, such as 8-oxoguanine (8oxoG), which impairs their function. OGG1 and MUTYH glycosylases initiate base excision repair (BER) at 8oxoG lesions. This study investigates their roles in telomeric 8oxoG-induced cellular senescence. Loss or inhibition of OGG1 or MUTYH partially rescues premature senescence and associated proinflammatory responses, while simultaneous deficiency in both glycosylases nearly completely prevents these effects. Additionally, glycosylase deficiency mitigates 8oxoG-induced telomere fragility and dysfunction, implicating single-stranded break (SSB) repair intermediates in impairing telomere replication.

Using a chemoptogenetic tool, researchers selectively induced 8oxoG production at telomeres in human fibroblasts deficient in OGG1, MUTYH, or both glycosylases. Single knockouts of OGG1 or MUTYH partially rescued hallmarks of telomeric 8oxoG-induced senescence, with reduced fragile telomeres compared to wild-type, and minimal increase observed in double knockout cells. Furthermore, preventing BER initiation in OGG1 and MUTYH-deficient cells suppressed telomeric 8oxoG-induced PARylation and rendered cells insensitive to the synergistic effects of PARP inhibitors on senescence induction.

Collectively, these findings suggest that incomplete BER at telomeres triggers cellular senescence through the accumulation of repair intermediates that disrupt telomere replication and stability.

General Comments:

1. While the study demonstrates the rescue of telomeric 8oxoG-induced senescence in OGG1 and MUTYH-deficient cells, more detailed mechanistic insights into how these glycosylases specifically influence cellular responses to 8oxoG lesions at telomeres would strengthen the impact of the findings. The authors could specifically discuss this aspect based on available literature if feasible.

2. It would also be insightful to discuss how the findings align with or diverge from existing literature on BER at telomeres and its implications for cellular senescence.

3. The study hints at potential therapeutic strategies involving PARP inhibitors in the context of BER deficiency. Expanding on the clinical relevance of these findings and discussing potential translational applications could broaden the impact of the study in the field of aging-related diseases or therapies targeting DNA repair pathways.

Specific comments:

1. In addition to β -galactosidase (β -gal) staining, incorporating additional senescence markers such as p16INK4a expression levels or telomere dysfunction-induced foci (TIFs) would strengthen the observation of cellular senescence induced by telomeric 8oxoG and glycosylase deficiency.

2. The study measures proinflammatory cytokines and chemokines secreted by senescent cells (SASP) in response to telomeric 8oxoG induction. Control experiments show that glycosylase-deficient cells exhibit reduced secretion of these

factors compared to wild-type (WT) cells (Fig. 2G). However, additional controls assessing basal levels of these factors in untreated conditions would strengthen the interpretation of DL-induced changes.

3. The observation of higher basal levels of certain proinflammatory factors in DKO cells, which do not increase further with DL exposure (Supplementary Fig. 2E), raises interesting questions about the baseline inflammatory state in glycosylase-deficient cells. Further investigation into the underlying mechanisms driving these observations would provide valuable insights.
4. While the study observes fewer fragile telomeres in glycosylase-deficient cells compared to WT after telomeric 8oxoG damage (Fig. 3B), providing quantitative data (e.g., percentage of cells with fragile telomeres) would enhance the robustness of the findings and support statistical analyses.
5. The study examines activation of the ATR/Chk1 and ATM/Chk2 pathways and downstream induction of p53 and p21 after telomeric 8oxoG damage (Fig. 3C, Supplementary Fig. 3C-E). It notes attenuated Chk1 phosphorylation in glycosylase-deficient cells and suggests less replication stress compared to WT. Including additional time points for DDR pathway activation (e.g., earlier time points post-DL exposure) could provide insights into the kinetics of DDR activation and its regulation by OGG1 and MUTYH. Furthermore, to strengthen the observation of DDR attenuation in glycosylase-deficient cells, additional controls such as mock-treated cells or cells treated with a specific ATR or ATM inhibitor would confirm the specific roles of these kinases in the observed responses.
6. The authors investigate the synergistic effects of Olaparib, a PARP inhibitor, on senescence induction after telomeric 8oxoG damage (Fig. 4F). To strengthen this observation, including quantitative data on cell viability or senescence markers (e.g., β -galactosidase staining) would provide a more comprehensive assessment of Olaparib's impact on glycosylase-deficient cells.
7. The study observes reduced PARylation levels at telomeres in quiescent cells compared to replicating cells after telomeric 8oxoG damage (Fig. 5B). Including additional controls, such as untreated cells or cells treated with a PARP inhibitor, would confirm the specificity of PARylation induction in response to telomeric 8oxoG and validate the impact of cell cycle status on PARylation dynamics.
8. IF-teloFISH analysis demonstrates telomere-specific PAR enrichment after telomeric 8oxoG damage (Fig. 5C, D). To further validate this finding, if technically feasible, performing co-localization studies with markers of DNA damage response (e.g., γ H2AX or 53BP1) would confirm the association of PARylation with damaged telomeres and provide mechanistic insights into the role of PARPs in response to 8oxoG-induced damage.
9. The study proposes that replication stress induced by BER intermediates enhances PARylation in response to telomeric 8oxoG damage (Fig. 5). To support this hypothesis, including experiments with inhibitors of DNA replication or BER enzymes would elucidate the specific contributions of replication stress and BER to PARylation and DDR activation.
10. While the study suggests that replication stress and BER contribute to 8oxoG-induced damage responses, integrating these findings with cell cycle-dependent outcomes (e.g., cell cycle arrest or apoptosis) would provide a comprehensive understanding of how different repair pathways modulate cellular responses to oxidative damage at telomeres.
11. The study reports an increase in cytoplasmic DNA species in DKO cells after chronic telomere damage, contrasting with the lack of increase after acute damage (Fig. 7A). Providing quantitative data, such as the percentage of cells with cytoplasmic DNA or the number of DNA fragments per cell, would strengthen this observation and support statistical analyses. Additionally, correlating these findings with other senescence markers (e.g., SA- β -gal staining) would provide a comprehensive assessment of cellular responses to chronic telomere damage.
12. The observation of increased chromatin bridges in DKO cells 24 hours after chronic telomere damage (Fig. 7B, C) is significant. However, including additional controls (e.g., untreated cells or cells treated with a known inducer of chromatin bridges) would confirm the specificity of this phenotype. Moreover, performing co-localization studies with DNA damage response markers (e.g., 53BP1) would validate the association of chromatin bridges with chromosome breaks and further elucidate the mechanism underlying genome instability in DKO cells.
13. The study implicates SSBs and MUTYH in telomeric 8oxoG-induced p53 activation (Supplementary Fig. 7C). To strengthen this finding, providing a time-course analysis of p53 expression post-damage and comparing it with other DNA repair-deficient cell lines would validate the specificity of p53 activation in response to telomeric 8oxoG-induced damage.

Version 1:

Reviewer comments:

Reviewer #1

(Remarks to the Author)

I would argue that the authors have effectively addressed the concerns raised.

Reviewer #2

(Remarks to the Author)

I went through the response to reviewers for my comments and I am satisfied that all of my concerns have been addressed.

Reviewer #3

(Remarks to the Author)

The authors have revised the manuscript thoroughly and significantly improved the manuscript. Additional data have strengthened the study. I recommend its acceptance.

We thank the reviewers for their time and effort in providing insightful comments on our study. To address these comments we have conducted new experiments, included additional data and controls, as well as clarified and revised the text, which we believe strengthens the manuscript. We highlight changes, additions, and points of clarification in red throughout the manuscript, and hope the reviewers agree that these revisions now make the manuscript suitable for publication.

Reviewer 1:

In this report, the authors used their chemoptogenetic tool (FAP-TRF1) expressed in BJ hTERT cells to evaluate the impact of oxidative damage to telomeres. Further, they evaluated the response as regulated by the base excision repair proteins OGG1 and MUTYH, two glycosylases that are selective for oxidative DNA damage. OGG1 removes the oxidative base lesion 8oxoG while MUTYH removes the A base opposite the oxidative base lesion 8oxoG. That removal (repair) process initiates base excision repair (BER) that could lead to the accumulation of DNA breaks and elevated PARP1/PARP2 activation if BER is not completed. From these studies they suggest that BER failure of such lesions at telomeres leads to the accumulation of single strand breaks that can initiate senescence and disrupt telomere function.

This is a well-designed and detailed study but some major concerns include:

1) The study is limited to a single cell line (human BJ hTERT fibroblasts expressing FAP-TRF1; BJ FAP-TRF1).

We published previously that telomeric 8oxoG induces senescence and replication stress in both RPE1-hTERT and BJ-hTERT cells ¹. Here we focused on genetically modifying BJ-hTERT cells to generate multiple cell lines: WT, OGG1 KO, MUTYH KO and OGG1, MUTYH double KO. We now acknowledge this limitation in our study and included “human fibroblasts” in our title and abstract.

In the Discussion we also now reference more studies in the literature that support our findings that OGG1-initiated BER can be detrimental since it increases senescence and breaks caused by 8-oxodGTP in A549 cancer cells ², H₂O₂-induced PARP1 overactivation in MEFs cells ³, and H₂O₂-induced telomere breaks in HeLa cells ⁴. These studies underscore that cell type and context influence the impact of glycosylase-induced repair intermediates.

As requested, we began repeating our study in the RPE1-hTERT FAP-TRF1 cells since different cell types might respond differently based on previous reports. We were not able to obtain bulk population KO cell lines in RPE FAP-TRF1, therefore we decided to conduct single cell cloning after CRISPR-Cas9 disruption of OGG1 and MUTYH. In the interest of a timely resubmission of this manuscript, we focused directly on the difference between WT and several DKO RPE FAP-TRF1 clones for rigor. We are excited to share with the reviewers that we observed rescue of telomeric 8oxoG-induced cell growth reduction in OGG1 and MUTYH double KO RPE1-hTERT clones treated with dye+light for 20 min and recovered for 4 days. We included the data below for the reviewers. This rescue was less pronounced than in the BJ FAP-TRF1 cells, and showed some variation among different clones in the extent of rescue and FAP-TRF1 expression. The greater sensitivity of repair-deficient RPE1-hTERT clones, compared to repair-deficient BJ-hTERT cells, may be due to a greater effect of the 8oxoG lesion itself on replication stress, which is not repaired in DKO cells. This increased sensitivity may be related to reports of higher metabolic activity causing higher ROS levels in RPE cells ⁵. We are in the process of testing several different single-cell clones of both OGG1 and MUTYH single KO that we also obtained, together with additional DKO clones. However, these studies are beyond the

scope of the current manuscript, and will require considerable time since multiple clones need to be analyzed for each genotype due to alterations that can arise during the cloning process. We added in the Discussion that cell type and context likely influence the response to glycosylase-

[Redacted]

2) The study is limited to a single KO for OGG1, a single KO for MUTYH and one double KO. It would be expected that a second set of KOs was developed to confirm some of the key findings, and also in at least a second cell line

We apologize for the misunderstanding. We clarified in the methods that these BJ-hTERT cells are not clonal cell lines, but rather are bulk populations obtained through CRISPR-Cas9 and targeting gRNAs. We share the concern that possible compensatory changes during single cell cloning can confound the results, therefore for the RPE FAP-TRF1 DKO cells we analyzed several different clones (see response to comment above). However, in our study for BJ-hTERT cells, we reproduced results similar to those obtained in the knockout cell lines, both with pharmacological inhibition of OGG1 and siRNA targeting OGG1. Moreover, for robustness, we now added also experiments with siRNA targeting MUTYH (Supplementary Fig. 4G-I). We reference this new data in the text when describing targeting siRNAs to deplete APE1 and OGG1.

3) It is noted that the KO cells were not complemented with an expression vector for OGG1 or MUTYH. It would be expected that such complemented cells would be developed to confirm the reversal of some of the key findings.

Unfortunately, the overexpression of glycosylases, including OGG1, has been published to be detrimental due to an imbalance in BER^{3,6,7}. This leads to excessive production of repair intermediates.

4) The authors show an increase in phospho-CHK1 after DL treatment in Figure 3 and then an increase in phospho-CHK2 in Figure S5. These markers represent different aspects of the DDR and so both should be evaluated in both datasets.

We thank the reviewer for pointing this out since phosphorylation of Chk1 is an important control for replication stress. We added phospho-Chk1 analysis to Supplementary Fig. 5A, and we indeed show that in quiescent cells there is no phospho-Chk1 after telomeric 8oxoG induction.

5) The almost opposing effects seen when comparing acute to chronic exposure is confounding. Can the authors either textually or experimentally expand on this aspect of the study?

The acute and chronic results are opposing for OGG1 KO and we now clarified possible reasons for this difference in the text. Our results are consistent with our previous study in HeLa cells that the accumulation of 8oxoG, which cannot be repaired in OGG1 KO cells, impairs growth. Based on published studies, we believe this is due to futile cycles of MUTYH or roles for MUTYH in signaling DNA damage, since we can partially rescue the impaired growth of OGG1 KO cells after chronic damage by also knocking out MUTYH (Figure 6B-C). We revised the Discussion to clarify our explanation.

6) The proposed model suggests that “chronic telomeric 8oxoG production revealed divergent roles for OGG1 and MUTYH in suppressing or promoting, respectively, damage-induced senescence or apoptosis” as depicted in Fig. 7D. This is confounding as both enzymes initiate BER in response to the 8oxoG base lesion. More explanation and maybe experimental analysis is needed to validate this claim.

We now expanded on this interesting difference in the Discussion and refer to data in the literature that indicates MUTYH has additional roles in DNA damage signaling beyond its glycosylase activity. Our results are consistent with findings in mouse models (reviewed in ⁸). Ogg1 knock out mice do not develop tumors even after KBrO₃ treatment despite increased G to T mutations ⁹. In contrast, Mutyh knock out mice show increased spontaneous and KBrO₃ induced intestinal tumors and G to T mutations ¹⁰. The difference in susceptibility to KBrO₃-induced tumorigenesis in Mutyh KO and Ogg1 KO has led to the suggestion that MUTHY has additional roles in tumor suppression, beyond preventing mutagenesis ^{8,11}.

Some minor concerns include: We clarified all the statistics.

Figure 1E – regarding the stats when comparing the OGG1-KO cells + DL to the DKO cells. + DL, nothing is indicated. Is that meant to suggest not significant?

Figure 1G – regarding the stats when comparing the OGG1-KO cells + DL to the DKO cells. + DL, nothing is indicated. Is that meant to suggest not significant?

Figure 2B – regarding the stats when comparing the OGG1-KO cells + DL to the DKO cells. + DL, nothing is indicated. Is that meant to suggest not significant?

Figure 2F – regarding the stats when comparing the OGG1-KO cells + DL to the DKO cells. + DL, nothing is indicated. Is that meant to suggest not significant?

Figure 2G – regarding the stats when comparing the OGG1-KO cells + DL to the DKO cells. + DL, nothing is indicated. Is that meant to suggest not significant?

Figure 3B – regarding the stats when comparing the OGG1-KO cells + DL to the DKO cells. + DL, nothing is indicated. Is that meant to suggest not significant?

Figure 4F – regarding the stats when comparing the OGG1-KO cells + DL + Olaparib to the DKO cells. + DL + Olaparib, nothing is indicated. Is that meant to suggest not significant?

Reviewer #2 (Remarks to the Author):

In this manuscript the authors describe results that support the conclusion that glycosylase activity on oxidized bases in the telomeres elicits toxic effects such as diminished cell proliferation in the presence of oxidized purines. The concept that repair intermediates can be worse than the offending base is consistent with orthogonal studies of AAG-driven BER imbalance. Importantly, this manuscript takes this paradigm a step further, pointing to the critical role of BER at telomeres as a modulator of cell survival, senescence, and stress responses. The results from these studies have important implications, since telomere stability plays such an important role in modulating phenotypes associated with diseases and aging. A remarkable finding is how powerful DNA damage at telomeres can be at driving biological responses, despite their being relatively small compared to the entire genome. It is noteworthy that the approach of looking at single and double KOs is elegant because the single KOs won't have opposing SSBs, by definition given the sequence at telomers. The extent to which single strand breaks (without opposing single strand breaks) can trigger complex cellular phenotypes, including a DNA damage response and senescence is an important finding. Another insight is that the double KO OGG1^{-/-}/MUTYH^{-/-} mice are prone to increased mutations (prior work) while at the same time reducing toxicity (this manuscript) which would conspire to increase viable mutant cells, thus driving cancer. The manuscript is expertly written with deep analysis. The combination of research excellence and exciting impact make this manuscript appropriate for publication in Nature Communications.

Overall, the quality of the data is sound and the appropriate techniques have been employed. There are some more minor conclusions that might be adjusted per the recommendations below. The results of this work significantly advance the field as well as helping to explain important biological observations. Below, please find general comments as well as comments that are specific to particular statements/figures.

1. General: The fact that 8oxoG can only be on one strand is of vital importance to interpreting the results and should be made clear at the outset, possibly in the introduction. This is an important aspect of the overall experimental design that would be obvious to someone who studies telomeres, but less obvious to a more general audience.

We thank the reviewer for pointing this out. We agree and clarified this in the Introduction.

2. General: Of note, OGG1 removes more than just 8oxoG. This should be clarified.

We clarified in the Results that OGG1 can also remove FapyG. However, there are no reports that singlet oxygen induces FapyG. So, we do not believe this activity is relevant to our study. Furthermore, NEIL1 can also remove FapyG and we reported previously that NEIL1 is not recruited to telomeres after FAP-TRF1-MG2I activation to produce singlet oxygen¹². This supports that the singlet oxygen produced in the FAP-TRF1-MG2I system does not generate FapyG lesions.

3. General: A lot is resting on Figure 1B for which there are only three points for each condition. Given that the results for Figure 1B are not consistent with the results in Supp Fig. 1B, it is important that confidence be at its highest for Figure 1B. Specifically, it is not clear why at 5 min of exposure each of the single KOs do not look different from WT while in Figure 1B they do look to be different. For this reason, it is recommended that the results in Figure 1B be repeated such that there are twice the number of values (6 values rather than 3 values for each bar). Of note, in many figures, an additive effect of the DKO is not evident. For example, in Figure 2B the DKO does not look different from the singles. This is also true for Figure 1E. This needs to be addressed.

In the Results, we clarified that Fig 1B and Supp Fig 1B (now Supp Fig 1C) are different because we used different exposure times, and therefore, induced different amounts of damage. Increasing the exposure from 5 min to 20 min, and thus the amount of damage, amplified the difference between WT and SKO cells. We showed previously that longer exposures induced more 8oxoG¹³. We find that greater amounts of damage, and therefore, increased opportunity for glycosylase processing into repair intermediate, exacerbates the differential response between WT, SKO and DKO cells. We also now repeated the experiment in Fig 1B 3 more times, for a total of 6 replicates.

We now clarified in Results, that we did not expect an additive effect in the DKO because the glycosylases do not have the same activities, and are not redundant. Only OGG1 can remove 8oxoG, and MUTYH activity depends on A misinserted opposite 8oxoG.

4. General: On line 136 it is stated that telomeres are more sensitive than the bulk genome. However, the method for introducing oxidative damage to the telomeres likely leads to clustering of damage that would not necessarily be the case for the global genome. Close clustering can lead to BER intermediates on opposing strands, leading to DSBs. This likely is not the case for cells exposed to KBrO₃. Without knowing total number of lesions or lesions per kb, it is hard to interpret this comparison.

The reviewer makes an excellent point, but since telomeres only have Gs on one strand, we do not expect 8oxoG formation on opposing strands. However, activities of OGG1 and MUTYH in close proximity could lead to a DSB in theory if one 8oxoG is repaired by OGG1 and the other is left unrepaired allowing for A misinsertion. We argue that BER on opposing strands that induce DSB is unlikely in our system because we reported the appearance of gammaH2AX and 53BP1 at telomeres depends on replication¹.

However, we have now added an estimate of the lesion frequency to the text (Results 1, Paragraph 2). We published recently that 20 min dye and light induces an average of 3-10 8oxoGs per 37kb telomeres in U2OS cells, based on our assay converting 8oxoG to double strand breaks in telomere restriction fragments, and comparing percent telomere cleavage to treatments with 20 or 40 mM KBrO₃ (for which 8oxoG lesion frequency in the bulk genome has been measured)¹⁴. Given that BJ-hTERT telomeres average about 10 kb, we estimate 20 min dye and light induces about 0.8 to 2 lesions per telomere, suggesting that clustered lesions is unlikely. Multiplying by 92 telomeres, this yields about 76 to 184 8oxoGs total.

HPLC quantification of cellular 8oxoG lesions reported 40 mM KBrO₃ induces 3-4 8oxoG per 10⁶ G^{15,16}, and 20 mM KBrO₃ induces 8 8oxoG per 10⁶ Gs¹⁷. Assuming a near linear dose relationship, 5 mM KBrO₃ induces an estimated 0.5 to 2 8oxoG per 10⁶ G. The human diploid genome has 6 billion bp (12 billion bases) with an estimated 24% Gs, totaling 2.88 billion Gs. Therefore, we predict that 5 mM KBrO₃ produces 1,400 to 5,700 8oxoGs and 10 mM KBrO₃ produces 2,800 to 11,400 8oxoGs per cell; much more than that produced at telomeres with our chemoptogenetic tool.

5. General: The text suggests that cGAS-STING may be activated by CCFs and more so in the WT cells (line 168 in reference to Figure 2F). However, proper analysis of cGAS-STING activity requires more than just looking at pro-inflammatory cytokines, especially since not all of those examined are downstream of STING activation. Furthermore, it is not clear why CXCL1, FAS, GDF-15 and MMP3 are shown in the main figure while GM-CSF, IL-6, IL-7, IL-8, and IL-15 are in the supplement. Of note, IL-6 is downstream of STING and the results show the opposite of the main conclusion (e.g., the DKO has more of an induction of IL-6, not less). This is particularly relevant given the recent results by the Boulton laboratory showing that it is often the

case that DNA damage does not induce STING activation. Rather than drawing conclusions about STING activation, perhaps it would be more conservative to focus on which aspects of the results reflect pro-inflammatory responses.

We chose to include the cytokines that change in the main text, and those that do not in the supplement to make the dense graphs more reader-friendly. We could include them all on one graph if required. Moreover, we expected not to see IL-6 upregulation following DL treatment based on our previous results ¹, and the DKO show an increased basal level of IL-6 but no further increase in response to DL treatment.

We now included a W blot to show STING upregulation following DL treatment (Supplementary Fig. 2E). However, the purpose of this analysis was to examine another endpoint of senescence (the SASPs). The field still has much to learn regarding mechanisms of SASPs activation. We suspect that after 4-7 days in culture some cytoplasmic DNA species (CCFs) may break down and allow DNA access to the cGAS activation. Notably the MN examined in the recent publication from the Boulton laboratory (Takai et al, Mol Cell, 2024) were produced by DNA-damage-induced mitotic errors. In contrast, we published that the telomeric 8oxoG induced CCFs likely arise from chromatin blebbing associated with senescent cells and not mitotic division ^{1,18}. Therefore the mechanisms may differ.

6. General: Figure 5 could be significantly strengthened. It appears that PAR levels were only assessed in one experiment; quantification across multiple experiments is wanted. Furthermore, the difference between +FBS and -FBS for PARGi+DL is not immediately apparent by eye. There is a considerable amount of variability, and the magnitude of the effect is slight. It is suggested that either additional experiments be performed, or perhaps to omit this figure. Additionally, while the data in Figure 5D shows the results from three independent experiments, the number of values is greater than 6 for the DL, PARGi -FBS bar. Clarifying how the analysis was done would be helpful.

We have now included biological replicates for the PAR blot in Fig. 5B and added the quantification, now shown in Fig. 5C. For the PAR IF analysis, now Fig. 5E, the points on top of the whiskers were not meant to indicate the number of values, which is shown below as n=. We clarified how the analysis was done in the figure legend. We show a box-and-whiskers plot where the whiskers indicate the 5-95th percentile range, and those points above the 95th percentile whisker indicate outliers (there are naturally no outliers below the 5th percentile because no values can be less than zero PAR foci per nucleus).

7. General: It is noteworthy that there is reduced cell growth for the single knockouts but not so much in the DKO in both Supp Fig. 1F and Figure 6B.

Yes, this is very interesting. Although we culture these cells at 5%O₂, we passage them in the lab at 20% O₂. Therefore, we expect that the cells experience some background oxidative damage. As we mentioned in the discussion, we propose that MUTYH may have a role in sensing or signaling unrepaired 8oxoG lesions, leading to a reduction in growth which is alleviated when MUTYH is knocked out of OGG1 deficient cells. MUTYH single knock out cells may grow more slowly than DKO because the presence of OGG1 can produce repair intermediates (SSBs) which may impair growth.

8. General: The paragraph beginning with line 312 is a bit hard to follow. Confusion stems from the change in the definition of CCF to make it CCF/MN. There is also a statement about MN arising from chromosome breaks (line 324) which introduces a new variable not discussed in other parts of the paper and for which the data have no error bars (Supp Figure 7G). Also, the interpretation would be very different if in Figure 7A the data were plotted as relative increase, in which case the WT look to have a more sensitive response than the DKO, which would be more

in sync with the earlier results. Finally, Figure 7D could be improved if the “+” notation was done for both acute and chronic responses.

Thank you for the comment. We now conducted additional chronic exposure replicates in p53 KO cells, therefore we now show the cytoplasmic DNA species analysis (Supplementary Fig. 7G) with three data points. Since we cannot be certain of the mechanism by which each cytoplasmic species arises, for consistency we altered the x-axis to read “CCF”. We made this correction in Figure 7 as well. The reviewer is correct that the fold increase for WT cells is greater than the fold increase for DKO cells. We clarified the text by stating “Interestingly, while the DKO cells showed less β -gal staining indicative of senescence, they showed the highest level of cytoplasmic DNA basally and after chronic damage, although WT and OGG1 KO cells showed the greatest damage-induced increase”. We also revised the model as suggested.

9. General: It would be helpful if in the results section there was some explanation for why there was a change in the protocol from a single DL exposure to multiple exposures over time. What is the hypothesis that motivated the long-term exposure paradigm?

We clarified this in the Results (section title **OGG1 loss sensitized cells to chronic telomere damage while MUTYH loss promotes senescence**, paragraph 1). As we reported previously, the single exposure is meant to mimic an acute oxidant exposure, and the repeated exposures mimic chronic oxidative stress. Inhibiting OGG1 may be beneficial to treat an acute inflammatory condition, but would be detrimental as a long term treatment; based on our data.

10. Line 54. It is stated that 8oxoG is more likely to form on telomeres. This reviewer is confused as to why/how this is the case and the extent to which there is solid prior work supporting this conclusion.

We added “sequences” after repeat to clarify and included citations and a review. This is because the telomeres are G-rich and have runs of Gs. Guanine has the lowest reduction potential among the natural bases, making it the most susceptible to oxidation reactions, and is even more susceptible to oxidation when present in G runs¹⁹. Biochemical studies show TTAGGG sequences are preferred sites for iron binding and Fenton reactions, as well as 8-oxoG formation^{20,21}. Some studies show more 8-oxoGs in telomeres compared to minisatellite sequences or the *36B4* gene locus in human and mouse cells after oxidant treatments²²⁻²⁴.

11. Line75. This reviewer would like to know what the relationship is rather than just stating that there is a role.

We clarified the text to state that “Mutyh deficiency in mice and humans increases tumorigenesis, along with 8oxoG-induced G to T mutations, which is exacerbated by additional OGG1 loss in mice”.

12. Line 108: It is stated that the DKO is rescued more than the single KOs. The issue at hand is whether or not 73% relative cell number for OGG1 KO is significantly different from 85% for the DKO. Statistics should be performed before this conclusion can be stated. Certainly one can say there is a trend.

We added a graph (new Supplementary Fig. 1B) which shows the statistical analysis and clarifies the differences among the cell lines.

13. Line 123: In reference to Figure 1C, it is stated that OGG1 is different from WT. While statistically 43% is less than 51%, this is a very slight difference and likely would not hold up if the entire experiment were repeated three times more. As such, it should be made clear that while the results are statistically significant, the magnitude of the difference is slight.

We have clarified this in the text. We agree that the magnitude is small, which further contrasts the results with telomere specific damage.

14. Figure 2B: It would be helpful if something was said about the fact that the DKO is not different from the singles.

We added “Damage-induced CCFs levels are similar in the single and double knock out cells, but are not higher in DKO compared to untreated cells after 24 h (Fig. 2B)”.

15. Line 159: Are bridges the only way to get CCF? If not, then this statement needs to be adjusted to indicate that other mechanisms of CCF production had not been explored and could help to explain why the glycosylase KO cells have more CCF.

In contrast, bridges are not thought to lead to CCFs. Rather evidence suggests they arise from blebbing of the chromatin in senescent cells due to a reduction in lamin B1 expression (for review see ¹⁸). We clarified this in the text, and we reported previously that acute oxidative damage a telomeres in BJ-FAP cells does not increase chromatin bridges as indicated by live cell imaging ¹.

16. Supp Figure 3: It is interesting that the single knockouts do not show attenuation of pATM or pChk2 while the DKO does. This is inconsistent with Figure 1E. This is curious and warrants mention at least, if not discussion.

We mention this in the text, and added that ATM and ATR signaling can “trigger senescence if not resolved”. Our results are likely due to differences in the kinetics of DDR activation (pATM and pChk2 observed at 3h after recovery), and the downstream consequences of senescence (observed at 4 days after recovery). The DDR signaling can lead to different cellular outcomes depending on the extent of the damage, and if resolved will not trigger senescence. We observe damage induced senescence in about 15% of the single knock out cells compared to the WT cells (~30%) 4 days after recovery (Figure 1E), consistent with lower production of repair intermediates.

17. Figure 4C: The MUTYH results are somewhat confusing. While the suggestion that MUTYH might assist OGG1 is interesting, much of the manuscript would need to be adjusted in terms of interpretation if this is an important mechanism of action. Interestingly, if this is a dominant activity of MUTYH, that would help to explain why in so many of the results one does not see an additive effect in the DKO. This requires deeper thought and possible adjustment of conclusions in other parts of the manuscript.

We do not believe this is a dominant MUTYH activity because there is no significant increase in repair intermediates after telomere damage in OGG1 KO (which still have MUTYH), but there is a significant increase in MUTHY KO cells. It is interesting that MUTHY KO cells show less damage induced repair intermediates than WT cells, and we believe this is consistent with some evidence that MUTYH may enhance OGG1-mediated 8oxoG repair, which we mention ¹¹. We agree this may partly explain why DKO effects are not additive. However, we believe our interpretation of the data is consistent with the biochemistry studies that MUTYH cannot remove 8oxoG.

18. Figure 6B: Another way to look at the data is to plot the final values at 24 days. A rough estimate is: WT 25%, OGG1 14%, MUTYH 50%, DKO 54%.

Yes, we agree this is interesting. But we think slope is helpful because the cell lines have different growth rates; the SKO lines grow more slowly than WT and DKO.

19. Line 285. This reviewer doesn't see a 'plateau'. Perhaps omit this descriptor.

We adjusted the text to state “The WT and OGG1 KO cells were the most severely affected by repeated DL exposures as indicated by the shallow slopes of their growth curves, and near

plateau for the OGG1 KO, while the MUTYH KO cells showed reduced growth but surpass the WT cells over time (Fig. 6B,C)".

20. Figure 6G: There looks to be many more points for the WT than for the DKO. Since the spread will increase with increasing number of values, these should be made equal.

We added the n values. They are similar.

21. Figure 7A: Is this CCF per micronucleus? The axis appears different from Figure 2, which is a bit confusing. Perhaps the analysis of CCF metrics should be consistent between Figure 2 and Figure 7.

This is number of CCFs per 100 cells. We clarified and made the axis labels consistent.

22. Supp Figure 7C. The results for the DKO look oddly absolute. If this were an analysis of phospho-P53 (which is the preferred approach), this might make sense. However, the results here suggest that there is no p53 at all in the untreated cells, which is unexpected. It is politely suggested that this experiment be repeated.

We repeated this experiment as suggested. The WB we showed in the original submission was a short film exposure of total p53 to show the induction in DL (x18) treated cells compared to the untreated cells. From a longer exposure, p53 is visible also in the untreated DKO (uploaded as uncropped gel, replicate #1). However, when carrying out additional chronic experiment replicates, we also performed a replicate of this p53 W blot, which now also includes extracts from p53 KO cells as a negative control (new Supplementary Fig. 7C-D). The longer film exposure clearly shows the DKO untreated cells have p53. We added an explanation of these results also in the text.

23. Line 350: It is suggested that the results for the chronic exposure be discussed in a separate paragraph from the main results since the findings are not as solid and are at times contradictory. This separate paragraph can also lay out the hypothesis, as mentioned above.

We appreciate the suggestion, but prefer to summarize our key findings in the first paragraph of the Discussion which also includes a description of all parts of the Model in Figure 7D. We do separate our discussion of the findings after acute damage and chronic damage in the subsequent paragraphs of the Discussion.

24. Paragraph starting on line 355. Many references are missing. Several laboratories have done work on imbalanced BER. References are needed for the first and second sentences.

We now reference an important review on this topic for the first sentence⁶. The second sentence is followed up with specific examples from the literature. We also now include more references on OGG1-induced intermediates in addition to the previously referenced work on alkylation damage in the next paragraph. We tried to avoid writing the Discussion as a review of

the literature due to space limitations, but agree there has been a lot of important work in this area.

25. Line 373: It is odd that this paragraph implies that MPG is not the same enzyme as AAG, which it is.

Thank you for pointing this out. We clarified this in the text.

26. Line 391: It is not clear from the results that the double KO is significantly different from OGG1 in Figure 1B, which would mean softening this statement in the discussion.

We have now included the statistics, by adding the graph in Supplementary Figure 1B.

27. Line 400: If there are not published data showing that unproductive binding of MUTYH causes PARylation, this statement should be removed.

We think it is fair to speculate the non-productive binding of MUTYH to 8oxoG:C may possibly provoke replication fork stalling and PARylation, based on reports that PARylation acts as a sensor of replication stress²⁵. We added this reference.

28. Line 405: It is noteworthy that a lot of the biology happens in the absence of replication. There is still PARylation in serum starved cells (Figure 5B) and the results in Figure 5D are not entirely convincing. Perhaps temper the conclusion here or do additional experiments.

We have now included biological replicates for the PAR blot in Figure 5B and added the quantification, now shown in 5C. We also explained better how the analysis in Figure 5D (now 5E) has been performed, to respond to a previous comment of this reviewer.

29. Line 407: If plotted as relative to untreated, the results are quite different. This is how the data were presented in Figure 1B, which is a centrally important figure.

We assume the reviewer is referring to Fig 6B here. Since this is a chronic exposure, we reported the data as population doubling. All the cell lines show decreased PD at 24 days for DL treated cells compared to UT cells, however, the extent of growth impairment differs among the cell lines, as shown in Fig 6C.

30. Line 422 and Figure 7D: What is the definition of “Genomic Instability”?

We refer to genomic instability here as the cytoplasmic DNA species which we speculate arise from chromatin bridge breakage in the DKO cells, since we see bridges after chronic telomere damage in the DKO cells.

31. Line 432: It is important to put the results of this manuscript into the context of what is known via the studies by Istvan Boldogh.

We are uncertain as to which studies the reviewer is specifically referring to. Work from Istvan Boldogh has led to important discoveries related to roles for OGG1 at promoters of inflammatory genes, including NFkB. In this context, we cited²⁶, which shows an OGG1 inhibitor can suppress pro-inflammatory gene expression, on which Dr. Boldogh was a co-author. Our study does not examine OGG1 activity at gene promoters, but we suggest OGG1 activity might also promote inflammation through activation of the SASP by generating repair intermediates at telomeres which promote senescence (detailed in the Discussion). However, we also now cite³ as further evidence that OGG1-initiated BER can exacerbate deleterious effects of oxidative stress, on which Dr. Boldogh is a co-author.

32. Line 443: This doesn't seem to be the strongest final statement. There is not really great evidence that MUTYH is acting on persistent unrepaired 8oxoG since the results can be explained by repeated induction of 8oxoG, which is different from the concept that a lesion is

sitting there unrepaired for a significant duration. It would be helpful if the authors made it a bit more clear what are the major/novel impacts of this work.

Our summary statement of key findings is “In summary, our study demonstrates that repair intermediates arising at telomeres from OGG1 and MUTYH glycosylase activity impair telomere replication and activate PARylation and DDR at telomeres, contributing to 8oxoG-mediated cellular senescence.” We don’t think our results of p53 activation after chronic damage can be explained simply by repeated induction of telomeric 8oxoG since we see p53 downregulation in treated DKO cells. Therefore, we think MUTYH may have a role in p53 signaling when OGG1 is absent and the 8oxoG lesions accumulate over time, and added “possibly” to the statement.

33. Details:

Line 63. **Remained.**

Line 108: Define DL. **Done**

Line 116: reverse the order in the parenthesis to read Supp figure number before the main figure number. **Done**

Reviewer #3 (Remarks to the Author):

Telomeres are critical for maintaining chromosome stability but are susceptible to oxidative damage, such as 8-oxoguanine (8oxoG), which impairs their function. OGG1 and MUTYH glycosylases initiate base excision repair (BER) at 8oxoG lesions. This study investigates their roles in telomeric 8oxoG-induced cellular senescence. Loss or inhibition of OGG1 or MUTYH partially rescues premature senescence and associated proinflammatory responses, while simultaneous deficiency in both glycosylases nearly completely prevents these effects. Additionally, glycosylase deficiency mitigates 8oxoG-induced telomere fragility and dysfunction, implicating single-stranded break (SSB) repair intermediates in impairing telomere replication.

Using a chemoptogenetic tool, researchers selectively induced 8oxoG production at telomeres in human fibroblasts deficient in OGG1, MUTYH, or both glycosylases. Single knockouts of OGG1 or MUTYH partially rescued hallmarks of telomeric 8oxoG-induced senescence, with reduced fragile telomeres compared to wild-type, and minimal increase observed in double knockout cells. Furthermore, preventing BER initiation in OGG1 and MUTYH-deficient cells suppressed telomeric 8oxoG-induced PARylation and rendered cells insensitive to the synergistic effects of PARP inhibitors on senescence induction.

Collectively, these findings suggest that incomplete BER at telomeres triggers cellular senescence through the accumulation of repair intermediates that disrupt telomere replication and stability.

General Comments:

1. While the study demonstrates the rescue of telomeric 8oxoG-induced senescence in OGG1 and MUTYH-deficient cells, more detailed mechanistic insights into how these glycosylases specifically influence cellular responses to 8oxoG lesions at telomeres would strengthen the impact of the findings. The authors could specifically discuss this aspect based on available literature if feasible.

We now refined the model (Fig 7D) to better convey the mechanistic findings of our work for both acute and chronic damage. Here, we demonstrate that OGG1 and MUTYH glycosylase activity triggers senescence via a mechanism of producing repair intermediates that impair telomere replication and stability. Using our chemoptogenetic tool to selectively generate 8oxoG

lesions at telomeres in human fibroblasts singly or doubly deficient in OGG1 and MUTYH glycosylases, we observed a partial or near complete rescue, respectively, of multiple hallmarks of senescence. Our data support a mechanistic model (Fig. 7D) in which glycosylase-mediated production of SSB intermediates causes replication stress and DDR activation at telomeres, leading to p53-mediated senescence. Telomeric 8oxoG induction in DKO cells fails to induce SSBs and PARylation at repair intermediates. However, we also now note that the cellular outcome glycosylase generated SSBs in general, and at telomeres, is likely influenced by the cell type and context. From the literature OGG1-initiated BER can be detrimental since it increases senescence and breaks caused by 8-oxodGTP in A549 cells ², H₂O₂-induced PARP1 overactivity in MEFs ³, and H₂O₂-induced telomere breaks in HeLa cells ⁴. These studies underscore that cell type and context influence the impact of glycosylase-induced repair intermediates.

2. It would also be insightful to discuss how the findings align with or diverge from existing literature on BER at telomeres and its implications for cellular senescence.

Our study is innovative for investigating BER specifically at telomeres, and in the context of non-cancer cell lines with fully intact DNA damage response pathways. Previous studies have reported MUTYH is recruited to oxidatively damaged telomeres in MEFs, but did not examine telomere integrity (losses or fragility) or senescence ²⁷. We reference our previous study that showed OGG1 is critical to telomere integrity after chronic telomere damage in HeLa cells ¹², which is consistent with what we found in OGG1 KO BJ-hTERT cells after chronic telomere damage here (Figs. 6H and 6I) (now highlighted in red). However, HeLa cells did not undergo senescence, likely because they are deficient in p53. Another study found that OGG1 inhibition exacerbated hydrogen peroxide and methotrexate induced telomere losses and micronuclei in U2OS cancer cells ²⁴, but senescence was not examined. This may suggest that OGG1 loss in cancer cells is detrimental to telomeres when damage is extensive, similar to our results in HeLa cells after chronic telomere oxidative damage. However, the Baquero et al. study involved oxidants that damage the entire genome, therefore, telomere losses may have arisen from replication fork collapse or double strand breaks arising between the centromere and telomere. Therefore, the cellular outcome cannot be attributed directly to BER events at the telomeres.

3. The study hints at potential therapeutic strategies involving PARP inhibitors in the context of BER deficiency. Expanding on the clinical relevance of these findings and discussing potential translational applications could broaden the impact of the study in the field of aging-related diseases or therapies targeting DNA repair pathways.

Thank you. We agree that our findings have interesting implications for translation, but we wished to limit our speculation due to space constraints. However, we would like to write a follow-up perspective piece. In the Discussion, we wrote “These findings raise the possibility that the combined use of conventional chemotherapeutic agents, which cause oxidative stress in normal tissues ²⁸, and PARP inhibitors, may contribute to premature cellular senescence and aging in cancer patients through a telomere-mediated mechanism.” In the last paragraph of the Discussion, we speculate on how OGG1 inhibitors which are being pursued as anti-inflammatory drugs, might also help suppress acute inflammation by preventing senescence and therefore, the associated SASP.

Specific comments:

1. In addition to β -galactosidase (β -gal) staining, incorporating additional senescence markers such as p16INK4a expression levels or telomere dysfunction-induced foci (TIFs) would strengthen the observation of cellular senescence induced by telomeric 8oxoG and glycosylase deficiency.

We showed previously that p16 is not induced by telomere damage, and p16 knock out does not rescue the telomeric 8oxoG induced senescence¹. We agree that several hallmarks of senescence are important to examine. Therefore we tested for DDR+ telomeres (TIFs), p21, p53, SASP, CCF, nuclear size increase, growth reduction and beta-gal staining. However, we have expanded the TIF analysis to show the number of actual TIFs per cell, (now shown in Supplementary Fig. 3G-H). These results recapitulate what is shown in Fig. 3E, but give more detailed information about the absolute number of γ H2AX⁺ and 53BP1⁺ telomeres per cell.

2. The study measures proinflammatory cytokines and chemokines secreted by senescent cells (SASP) in response to telomeric 8oxoG induction. Control experiments show that glycosylase-deficient cells exhibit reduced secretion of these factors compared to wild-type (WT) cells (Fig. 2G). However, additional controls assessing basal levels of these factors in untreated conditions would strengthen the interpretation of DL-induced changes.

The SASP data for untreated cells has been included. The untreated cells were not treated with dye and light, and represent basal levels. We now highlighted this in the text.

3. The observation of higher basal levels of certain proinflammatory factors in DKO cells, which do not increase further with DL exposure (Supplementary Fig. 2E), raises interesting questions about the baseline inflammatory state in glycosylase-deficient cells. Further investigation into the underlying mechanisms driving these observations would provide valuable insights.

We agree, but exploring the underlying mechanism for the increase in inflammatory response is beyond the scope of this paper. The purpose here was to examine SASP, an important marker of senescence, which is considered a gold standard for defining a senescent cell.

4. While the study observes fewer fragile telomeres in glycosylase-deficient cells compared to WT after telomeric 8oxoG damage (Fig. 3B), providing quantitative data (e.g., percentage of cells with fragile telomeres) would enhance the robustness of the findings and support statistical analyses.

For this analysis, we followed the standard in the field for ease of comparison with the published studies which typically report the data as number of fragile telomeres per metaphase or % fragile telomeres per metaphase (see for example the publication originally describing fragile telomere²⁹). Each data point represents an analyzed metaphase, which represents a cell. The number of fragile telomeres in each cell, for instance, offers a clear view of the extent of replication stress occurring at multiple chromosome ends in each analyzed cell (i.e. data point).

5. The study examines activation of the ATR/Chk1 and ATM/Chk2 pathways and downstream induction of p53 and p21 after telomeric 8oxoG damage (Fig. 3C, Supplementary Fig. 3C-E). It notes attenuated Chk1 phosphorylation in glycosylase-deficient cells and suggests less replication stress compared to WT. Including additional time points for DDR pathway activation (e.g., earlier time points post-DL exposure) could provide insights into the kinetics of DDR activation and its regulation by OGG1 and MUTYH. Furthermore, to strengthen the observation of DDR attenuation in glycosylase-deficient cells, additional controls such as mock-treated cells or cells treated with a specific ATR or ATM inhibitor would confirm the specific roles of these kinases in the observed responses.

We published on earlier time points (30, 60 and 180 min) and selected 180 min (3 hr) since this is when we observed maximal response in the WT cells¹. We agree that ATM and ATR inhibitors are interesting. We published that ATM inhibitor suppresses the telomeric 8oxoG induced senescence¹. We also have interesting findings that ATRi increases sensitivity to telomeric 8oxoG induced senescence, and causes a stimulation in MN formation. This is part of a larger study focused on ATR signaling and telomeric 8oxoG, and the manuscript is currently in preparation with Dr. Barnes (KUMC) as co-corresponding author.

6. The authors investigate the synergistic effects of Olaparib, a PARP inhibitor, on senescence induction after telomeric 8oxoG damage (Fig. 4F). To strengthen this observation, including quantitative data on cell viability or senescence markers (e.g., β -galactosidase staining) would provide a more comprehensive assessment of Olaparib's impact on glycosylase-deficient cells. We agree and include data on beta-galactosidase staining as a marker of senescence in Fig. 4F, showing that DL and PARPi treatments have a synergistic effect on cellular senescence induction in repair proficient cells.

7. The study observes reduced PARylation levels at telomeres in quiescent cells compared to replicating cells after telomeric 8oxoG damage (Fig. 5B). Including additional controls, such as untreated cells or cells treated with a PARP inhibitor, would confirm the specificity of PARylation induction in response to telomeric 8oxoG and validate the impact of cell cycle status on PARylation dynamics.

We showed that there is no PARylation in UT cells (-) (Fig. 4D, lanes 1,3,5,7 and Fig. 5B, lanes 1,3). We tested for PARylation after treatment with the PARPi Olaparib in Supplementary Fig. 4E (lanes 7,8,9). We performed this control experiment in unsynchronized cells, but we expect no PARylation if we treated quiescent cells (-FBS) as well with Olaparib and tested for PAR. However, we have now added new quantification and statistical analysis for Fig. 5B that shows the extent of PARylation reduction in the quiescent cells compared to the unsynchronized (+FBS), which is in the new Fig. 5C.

8. IF-teloFISH analysis demonstrates telomere-specific PAR enrichment after telomeric 8oxoG damage (Fig. 5C, D). To further validate this finding, if technically feasible, performing co-localization studies with markers of DNA damage response (e.g., γ H2AX or 53BP1) would confirm the association of PARylation with damaged telomeres and provide mechanistic insights into the role of PARPs in response to 8oxoG-induced damage.

Our evidence suggests the gammaH2AX and 53BP1 activation are downstream of PARylation since these foci arise as a consequence of impaired telomere replication. We showed previously that gammaH2AX is not significantly induced after telomere damage in quiescent cells¹. Technically this experiment would be very challenging because PAR foci analysis must be done right after damage in PARGi treated cells to preserve the PAR chains. Long term treatment with PARGi is toxic. But we analyze gammaH2AX and 53BP1 foci at telomeres after a 24 h recovery, to allow for replication because they are induced by replication stress.

9. The study proposes that replication stress induced by BER intermediates enhances PARylation in response to telomeric 8oxoG damage (Fig. 5). To support this hypothesis, including experiments with inhibitors of DNA replication or BER enzymes would elucidate the specific contributions of replication stress and BER to PARylation and DDR activation.

We chose to inhibit replication by serum starvation and observed decreased PARylation compared to asynchronous replicating cells (Fig. 5B). Chemical replication inhibitors (HU, aphidicolin) can induce DNA damage and telomere fragility, this is why we chose to induce quiescence with serum starvation, which does not induce telomere fragility. We now show telomeric 8oxoG induces phospho-Chk2, but not phospho-Chk1, in quiescent non-replicating cells (Supplementary Fig. 5A). This indicates that BER contributes to DDR through Chk2 activation even in non-replicating cells. Whereas, replication stress contributed to the telomeric 8oxoG-induced Chk1 activation, since this does not occur in quiescent cells. DKO cells show greatly suppressed PARylation (Fig. 4D), Chk1 phosphorylation and Chk2 phosphorylation (Fig. 3C and Supplementary Fig. 3C), as well as no SSB repair intermediates (Fig. 4C), indicating that repair intermediates (i.e. BER activity) contribute to PARylation, DDR signaling, and replication stress-induced Chk1 activation.

10. While the study suggests that replication stress and BER contribute to 8oxoG-induced damage responses, integrating these findings with cell cycle-dependent outcomes (e.g., cell cycle arrest or apoptosis) would provide a comprehensive understanding of how different repair pathways modulate cellular responses to oxidative damage at telomeres.

As mentioned above, we chose to induce cell cycle arrest by serum starvation, since chemical agents that induce cell cycle arrest also cause telomere fragility. We now show that telomeric 8oxoG induced Chk1 phosphorylation requires replication and cell cycle progression, whereas Chk2 phosphorylation does not (Supplementary Figure 5A). We published previously, that telomeric 8oxoG-induced cellular senescence is dependent on replication and cell cycle progression, as well as p53 activation¹. We only observe apoptosis after chronic telomere damage in WT and OGG1 KO cells, but not after acute damage in any of the cell lines (Supplementary Fig 6A, and Supplementary Fig 2C-D). We agree that “understanding how different repair pathways modulate the cellular response to oxidative damage at telomeres”, is very interesting and we are designing CRISPR-Cas9 screens with genome wide and targeted gRNA libraries for this. But these experiments are beyond the scope of the current study which is focused on OGG1 and MUTYH, as noted in the title.

11. The study reports an increase in cytoplasmic DNA species in DKO cells after chronic telomere damage, contrasting with the lack of increase after acute damage (Fig. 7A). Providing quantitative data, such as the percentage of cells with cytoplasmic DNA or the number of DNA fragments per cell, would strengthen this observation and support statistical analyses.

Additionally, correlating these findings with other senescence markers (e.g., SA- β -gal staining) would provide a comprehensive assessment of cellular responses to chronic telomere damage. We included SA- β -gal staining as a marker of senescence after chronic in Figure 6D. As for correlating the CCFs with SA- β -gal directly, we have tried to stain for DAPI and β -gal at the same time, but the two experimental protocols are not compatible with each other. We cannot assign CCF to a particular cell without being able to visualize the cytoplasm, and this is why we followed the standard of CCFs per 100 cells to be conservative.

12. The observation of increased chromatin bridges in DKO cells 24 hours after chronic telomere damage (Fig. 7B, C) is significant. However, including additional controls (e.g., untreated cells or cells treated with a known inducer of chromatin bridges) would confirm the specificity of this phenotype. Moreover, performing co-localization studies with DNA damage response markers (e.g., 53BP1) would validate the association of chromatin bridges with chromosome breaks and further elucidate the mechanism underlying genome instability in DKO cells.

Unfortunately, 53BP1 foci do not form on chromatid bridges. However, we now acknowledge in the text that we can only infer bridges lead to breaks, but we cannot prove this. This requires live cell imaging, these events would be exceedingly difficult to follow because they are very rare at ~1.5 per 100 cells.

13. The study implicates SSBs and MUTYH in telomeric 8oxoG-induced p53 activation (Supplementary Fig. 7C). To strengthen this finding, providing a time-course analysis of p53 expression post-damage and comparing it with other DNA repair-deficient cell lines would validate the specificity of p53 activation in response to telomeric 8oxoG-induced damage.

We previously validated the specificity of p53 for telomeric 8oxoG induced damage by showing complete rescue of telomeric 8oxoG induced senescence in p53 KO cells¹. Similarly to the acute treatment experiments, we chose to examine the time point after damage that gave the highest induction of p53 in the WT cells (3h post damage for acute and 3h after the last DL chronic treatment, for consistency), and in both these experiments we compare the WT cells to

the repair deficient cells. As mentioned above we are conducting a targeted CRISPR-Cas9 screen for DNA damage response and replication response factors, but this is outside the scope of the current manuscript. However, to address specificity we have treated cells with another DNA damaging agent, UV irradiation, and found that p53 is activated in the DKO cells (Supplementary Fig. 7D).

To further strengthen this result, we now repeated this chronic experiment for an additional independent biological replicate, and included the p53 KO BJ FAP TRF1 cells as a negative control. The revised blots are now in Supplementary Figs. 7C-D. Please see also response to Review 2, comment 22.

A time course would be challenging because these cells are treated repeatedly over the course of a month, and we do expect this would provide significant insight into the mechanism of p53 suppression. Our surprising finding in DKO cells has initiated a new project in the lab to decipher the mechanism of p53 suppression. We plan to conduct mRNA-Seq experiments, and will follow-up on the top hits. We are also considering mRNA-Seq at various time points during the one month exposure, depending on the expense. This would give us unbiased information on differences in the DDR response in WT versus glycosylase deficient cells, and we will compare to different DNA damaging agents. We anticipate these studies could open a new line of inquiry, for a future study.

1. Barnes, R.P. et al. Telomeric 8-oxo-guanine drives rapid premature senescence in the absence of telomere shortening. *Nat Struct Mol Biol* **29**, 639-652 (2022).
2. Zhang, L. et al. OGG1 co-inhibition antagonizes the tumor-inhibitory effects of targeting MTH1. *Redox Biol* **40**, 101848 (2021).
3. Wang, R. et al. OGG1-initiated base excision repair exacerbates oxidative stress-induced parthanatos. *Cell Death Dis* **9**, 628 (2018).
4. Ahmed, W. & Lingner, J. PRDX1 Counteracts Catastrophic Telomeric Cleavage Events That Are Triggered by DNA Repair Activities Post Oxidative Damage. *Cell Rep* **33**, 108347 (2020).
5. Jager, R.D., Mieler, W.F. & Miller, J.W. Age-related macular degeneration. *N Engl J Med* **358**, 2606-17 (2008).
6. Fu, D., Calvo, J.A. & Samson, L.D. Balancing repair and tolerance of DNA damage caused by alkylating agents. *Nat Rev Cancer* **12**, 104-20 (2012).
7. Ebrahimkhani, M.R. et al. Aag-initiated base excision repair promotes ischemia reperfusion injury in liver, brain, and kidney. *Proc Natl Acad Sci U S A* **111**, E4878-86 (2014).
8. Nakabeppu, Y. Cellular levels of 8-oxoguanine in either DNA or the nucleotide pool play pivotal roles in carcinogenesis and survival of cancer cells. *Int J Mol Sci* **15**, 12543-57 (2014).
9. Arai, T., Kelly, V.P., Minowa, O., Noda, T. & Nishimura, S. The study using wild-type and Ogg1 knockout mice exposed to potassium bromate shows no tumor induction despite an extensive accumulation of 8-hydroxyguanine in kidney DNA. *Toxicology* **221**, 179-86 (2006).
10. Sakamoto, K. et al. MUTYH-null mice are susceptible to spontaneous and oxidative stress induced intestinal tumorigenesis. *Cancer Res* **67**, 6599-604 (2007).

11. Raetz, A.G. & David, S.S. When you're strange: Unusual features of the MUTYH glycosylase and implications in cancer. *DNA Repair (Amst)* **80**, 16-25 (2019).
12. Fouquerel, E. et al. Targeted and Persistent 8-Oxoguanine Base Damage at Telomeres Promotes Telomere Loss and Crisis. *Mol Cell* **75**, 117-130 e6 (2019).
13. Barnes, R.P., Thosar, S.A., Fouquerel, E. & Opresko, P.L. Targeted Formation of 8-Oxoguanine in Telomeres. *Methods Mol Biol* **2444**, 141-159 (2022).
14. Thosar, S.A. et al. Oxidative guanine base damage plays a dual role in regulating productive ALT-associated homology-directed repair. *Cell Rep* **43**, 113656 (2024).
15. Parlanti, E. et al. The cross talk between pathways in the repair of 8-oxo-7,8-dihydroguanine in mouse and human cells. *Free Radic Biol Med* **53**, 2171-7 (2012).
16. De Luca, G. et al. A role for oxidized DNA precursors in Huntington's disease-like striatal neurodegeneration. *PLoS Genet* **4**, e1000266 (2008).
17. Visnes, T. et al. Targeting OGG1 arrests cancer cell proliferation by inducing replication stress. *Nucleic Acids Res* **48**, 12234-12251 (2020).
18. Miller, K.N. et al. Cytoplasmic DNA: sources, sensing, and role in aging and disease. *Cell* **184**, 5506-5526 (2021).
19. Fukuzumi, S., Miyao, H., Ohkubo, K. & Suenobu, T. Electron-transfer oxidation properties of DNA bases and DNA oligomers. *J Phys Chem A* **109**, 3285-94 (2005).
20. Henle, E.S. et al. Sequence-specific DNA cleavage by Fe²⁺-mediated fenton reactions has possible biological implications. *J Biol Chem* **274**, 962-71 (1999).
21. Oikawa, S., Tada-Oikawa, S. & Kawanishi, S. Site-specific DNA damage at the GGG sequence by UVA involves acceleration of telomere shortening. *Biochemistry* **40**, 4763-8 (2001).
22. O'Callaghan, N., Baack, N., Sharif, R. & Fenech, M. A qPCR-based assay to quantify oxidized guanine and other FPG-sensitive base lesions within telomeric DNA. *Biotechniques* **51**, 403-11 (2011).
23. Rhee, D.B., Ghosh, A., Lu, J., Bohr, V.A. & Liu, Y. Factors that influence telomeric oxidative base damage and repair by DNA glycosylase OGG1. *DNA Repair (Amst)* **10**, 34-44 (2011).
24. Baquero, J.M. et al. Small molecule inhibitor of OGG1 blocks oxidative DNA damage repair at telomeres and potentiates methotrexate anticancer effects. *Sci Rep* **11**, 3490 (2021).
25. Mosler, T. et al. PARP1 proximity proteomics reveals interaction partners at stressed replication forks. *Nucleic Acids Res* **50**, 11600-11618 (2022).
26. Visnes, T. et al. Small-molecule inhibitor of OGG1 suppresses proinflammatory gene expression and inflammation. *Science* **362**, 834-839 (2018).
27. Tan, J. et al. An ordered assembly of MYH glycosylase, SIRT6 protein deacetylase, and Rad9-Rad1-Hus1 checkpoint clamp at oxidatively damaged telomeres. *Aging (Albany NY)* **12**, 17761-17785 (2020).
28. Zhang, Y. et al. Chemotherapeutic drugs induce oxidative stress associated with DNA repair and metabolism modulation. *Life Sci* **289**, 120242 (2022).
29. Sfeir, A. et al. Mammalian telomeres resemble fragile sites and require TRF1 for efficient replication. *Cell* **138**, 90-103 (2009).

We thank the reviewers for their time and effort in reviewing our revised manuscript. We are delighted that their prior issues have been addressed to their satisfaction.